# Genetic algorithm-based coverage path planning for autonomous aircraft cabin cleaning by reconfigurable robot

**Cong Hien Dinh[1], Chong Yong Qi[2], Huynh Van Van[3], Guangming Chen[4], Rajesh Elara Mohan[2], Anh Vu Le[3]***

1 Faculty of Electrical and Electronics Engineering, Ton Duc Thang University, Ho Chi Minh City, Vietnam, 2 ROAR Lab, Engineering Product Development, Singapore University of Technology and Design, Singapore, Singapore, 3 Advanced Intelligent Technology Research Group, Faculty of Electrical and Electronics Engineering, Ton Duc Thang University, Ho Chi Minh City, Vietnam, 4 College of Mechanical and Electrical Engineering, Nanjing University of Aeronautics and Astronautics, Nanjing, People's Republic of China

* leanhvu@tdtu.edu.vn

## Abstract

Designing an optimal Coverage Path Planning (CPP) framework for autonomous aircraft cabin cleaning is a critical challenge due to the time-sensitive nature of aircraft turnaround operations. Conventional domestic cleaning robots struggle to adapt to the confined and irregular cabin layouts of commercial aircraft. To address this, the paper proposes a two-stage CPP approach utilizing the reconfigurable robot. In the first stage, the robot operates in its full-size configuration to efficiently clean open regions such as aisles and galleys, skipping hard-to-access seat rows to minimize total cleaning time. In the second stage, a Genetic Algorithm (GA)-based Traveling Salesman Problem (TSP) optimization process determines the optimal visiting sequence for the skipped areas, while simultaneously accounting for the robot's reconfiguration energy model. This integrated framework explicitly models the trade-off between coverage efficiency, energy consumption, and reconfiguration cost, ensuring that the robot autonomously selects the most energy-optimal path under operational constraints. The experiments incorporating airline procedures and cabin geometry demonstrate that the proposed approach significantly outperforms conventional CPP strategies in both coverage time and energy usage. The results validate the feasibility of deploying reconfigurable robotic systems for real-world autonomous aircraft cabin cleaning during turnaround operations.

## 1 Introduction

According to research conducted by the International Air Transport Association (IATA) in 2019, it was predicted that the number of global airline passengers could double by 2037. This projection is primarily attributed to rising standards of living across

**Data availability statement:** All relevant data are within the manuscript and its Supporting Information files.

**Funding:** The author(s) received no specific funding for this work.

**Competing interests:** The authors have declared that no competing interests exist.

the globe, increased accessibility to air travel due to lower transportation costs, the expansion of flight routes, and globalization [1]. Consequently, the surge in airline passengers signifies a growing demand for air travel. As a result, the increase in air travel gives rise to the need for continuous cleaning of the aircraft's interior, prompting questions about when, where, and how aircraft cabins are cleaned.

During each journey, whether at the origin or destination, passengers board and disembark from the aircraft. Once all passengers have disembarked, the operating airline seizes the opportunity to clean the aircraft as shown in Fig 1. This is a standard procedure in the aviation industry and is commonly referred to as an 'aircraft turnaround' [2]. Aircraft turnarounds typically involve maintenance and refueling. Cleaners are deployed to board the stationary aircraft and clean the interior of the cabin [3].

Aircraft cleaning is a unique occupation characterized by a high concentration of physical activities within limited time and space, which are not entirely under the direct control of service providers and their workers, as described in [4,5]. Additionally, airlines are increasingly advocating for denser seat layout configurations in their aircraft for passenger maximization. This move is driven by the necessity to accommodate more passengers on each flight. Consequently, commercial flight models demand very tight and constrained cleaning schedules. These factors have led to potential underlying risk factors for the development of disorders among cleaners, as emphasized in [5,6].

Utilizing autonomous processes that leverage advanced cleaning robots represents a viable solution for enhancing cleaning efficiency during aircraft turnarounds. However, the aircraft environment differs significantly from the robot's typical operating environment. Typical living spaces and environments, such as homes, offices, and public areas, have distinct layouts when compared to the aircraft cabin. While typical living spaces are open, the interior of the aircraft cabin can be characterized as a densely arranged and cluttered environment, featuring numerous hard-to-access spaces that require cleaning [7].

The objective of CPP is to determine an optimal path that ensures complete traversal of a specified area while meeting three essential criteria [8]. First, the path must guarantee full coverage, meaning that every point within the workspace is visited. Second, it should avoid repetition, ensuring that no point is visited more than once. Finally, the solution must satisfy quality requirements specific to the application, such as minimizing motion complexity or energy consumption.

CPP approaches find application in a wide range of scenarios where there is a need to maximize coverage in specific environments. These applications span diverse fields, such as agriculture [9], search and rescue emergencies in emergencies and natural disasters [10], operating in hazardous conditions like in oil and gas power plant ([11] and [12]), lawn mowing [13], and shiphull cleaning [14].

For cleaning robots, the primary objective of CPP is typically to achieve optimal coverage of the targeted cleaning environment. This objective can be approached through different methods. The first, which is a more straightforward approach, involves the robot relying on simple bump sensors to detect obstacles and make

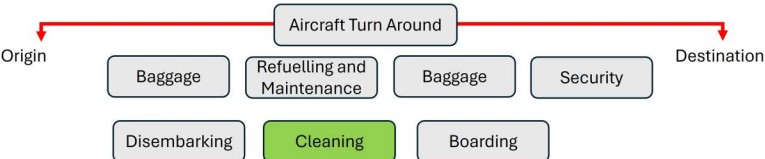

**Fig 1. The various sequence of operation of the aircraft, at either its Origin or Destination location, from the time it lands and throughout the entire turnaround stopover.**

route corrections. In these randomized coverage algorithms, cleaning robots are often simplistic and less efficient, lacking laser sensors for mapping to understand the environment. In contrast, the second approach equips the robot with more advanced positioning equipment like IR sensors, 2D/3D LIDAR, GPS, and a CCP algorithm, allowing for a more efficient cleaning path [15,16]. Hence, these diverse CPP algorithms are widely adopted in modern cleaning robots and can be broadly categorized as randomized and sensor-based coverage algorithms.

Domestic cleaning robots are rising in demand and are designed to navigate within complex homes or living space settings, typically without prior knowledge of the environment. They are tasked with creating a map of the environment and operating in dynamic settings where occupants and moving obstacles are presented. Therefore, this context requires a robust and adaptable approach, necessitating the assessment of relevant studies and references. Furthermore, cleaning robots in challenging environments must access hard-to-reach areas, such as under seats, where human cleaners often face difficulties (see Fig 2), which raises the need for various shape size robot or robot have the ability of shape-shifting.

Robotic path planning (PP) in a working environment can be categorized as static or dynamic. A static environment is one where obstacles do not move, and the environment is typically known. In contrast, a dynamic environment involves moving obstacles, requiring adaptability. In the context of an aircraft interior cabin, most fixtures, galleys, aircraft seats, and lavatories are bolted into the cabin and are therefore considered static obstacles [9,17] and [18].

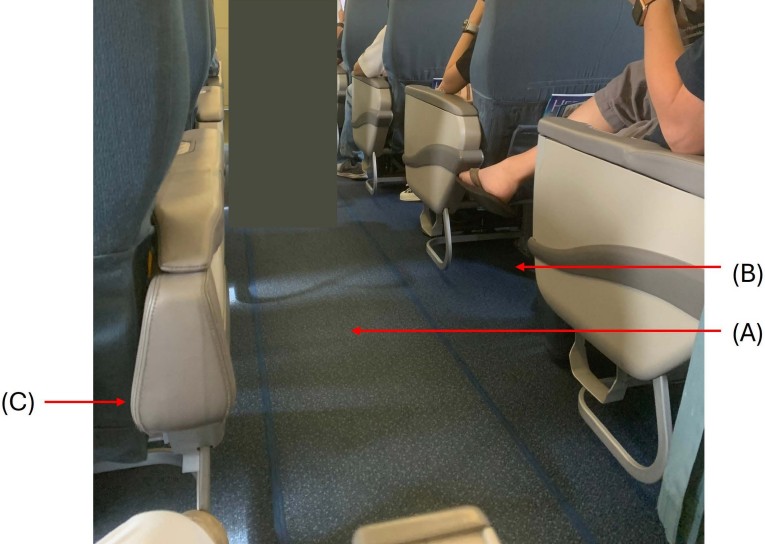

**Fig 2. All areas must be clean during aircraft turnaround.** A: Cleaning along the aisle. B: Cleaning in between seats. C:eaning in underneath seats.

Cleaning robots, whether equipped with simple bump sensors and a basic random coverage algorithm or advanced sensors and algorithms, often employ one or a combination of several cleaning patterns [19,20]. Common programmed CPP patterns, including zigzag, spiral, and wall-following, are not always sufficient. Introducing advanced cleaning robots and adapting CPP to them for cleaning within the confines of an aircraft cabin during a limited turnaround interval is a required study.

By solving the challenge of finding optimal cleaning trajectories, in this research, we will focus on adopting the combination of GA with TSP-based optimal path planning for the aircraft's interior cabin context cleaning. Hence, the primary focus is to clean the aircraft cabin interior during turnover by establishing a practical approach for interpreting the data collected from the cleaning robots. This paper intends to deploy the robot which can reconfigure the shape through the cleaning process using two distinct stages. In stage 1, the robot with the full size to speed up the cleaning time (note that the bigger the robot size, the faster it covers the space) will traverse the entire aircraft cabin by simple algorithm such as zigzag or spiral while bypassing rows of seats identified with issues, such as obstacles as hard to access space that could impede the robot's cleaning by current size. During this stage, the robot also records the problematic rows as waypoints by the perception module and sends these to the next stage. In stage 2, the robot adaptively reconfiguring to the smaller footprint size will employ a GA Optimization-based Traveling Salesman Problem (TSP) approach (one of advance technique as a form of Evolutionary Computation) to revisit only the rows with identified issues that were skipped during stage 1 in the optimal sequences. To validate the CPP approach, we utilize two base sizes of commercial Smorphi as a case study [21] as a reconfigurable system. Subsequently, the paper presents experimental results and discussions that validate the CPP approach, encompassing the time required for cleaning and the percentage of area coverage within the mapped environment. In this work, complete coverage is evaluated at the level of the integrated two-stage framework, rather than at the individual stage level.

The main contributions of this paper include an exploration of the feasibility of deploying cleaning reconfigurable robots with flexible base sizes in aircraft cabin interiors through GA-based optimal CPP. This approach consideration incorporates real-world operational procedures, including aircraft boarding and disembarking procedures with two real aircraft cabin environments, as well as passenger layout patterns, to assess the potential adoption of the proposed method.

The paper's objectives are as follows:

- Ability to reach far corners by reconfigurable robot with flexible size in the optimal time of the cabin and under the seats.

- Automate the aircraft cabin interior cleaning by CPP method and, in turn, provide a more efficient time required for cleaning during the aircraft turnaround by using the reconfigurable full and half-size robot footprint.

The structure of the paper is as follows. Section 2 presents the related work about PP and CPP and applies these approaches to aircraft cleaning. Section 3 describes a reconfigurable platform. Section 4 presents the proposed complete coverage path planning framework for aircraft cabin cleaning and experimental results. Section 5 presents the discussion and limitations. Section 6 gives the future work and summarizes the paper.

## 2 Related work

Coverage Path Planning (CPP) addresses the challenge of generating a trajectory that ensures every reachable area in a given environment is visited at least once, while also optimizing metrics such as travel distance, time, or energy consumption [13]. This problem is fundamental in applications such as robotic floor cleaning, agricultural coverage, surveillance, and industrial inspection, where the objective is not only to explore the environment but to exhaustively and efficiently cover all task-relevant regions.

While the aircraft cabin presents a highly structured and repetitive environment that is well-suited to offline path planning [9], real-world operations frequently involve unpredictable conditions. For instance, particularly dirty zones or high-touch surfaces may be detected dynamically using onboard perception systems, such as visual inspection or particulate

 

sensors [22,23]. These areas, which are not included in the initial offline map, are treated as dynamically generated waypoints requiring prioritized, targeted cleaning. Consequently, the coverage task transitions from a purely pre-planned operation into a hybrid planning problem that combines structured offline sweeping with an adaptive response to real-time demands [13].

To address this, adaptive partial Coverage Path Planning strategies are employed. These methods allow the robot to compute an optimized local path through the new waypoints without performing a full global replanning. This local optimization is often modeled as a variation of the Traveling Salesman Problem (TSP), and heuristic or evolutionary approaches such as Genetic Algorithms (GAs) have shown strong performance in solving such problems efficiently [24,25]. The integration of optization-based waypoint optimization into the overall CPP framework enables responsive, energy-efficient, and mission-adaptive operations. Thus, this hybrid strategy—merging overall coverage for general areas with adaptive planning for dynamically identified regions—ensures comprehensive and flexible cleaning performance in the complex and semi-dynamic environment of an aircraft cabin.

Several classical approaches have been applied to make the structured environments. Cell decomposition techniques divide the space into smaller regions, such as grids or trapezoidal cells, and then apply systematic sweeping patterns such as boustrophedon motions. These methods are relatively simple and scalable but often produce suboptimal global paths due to unnecessary backtracking when transitioning between subregions [26]. Spanning Tree Coverage (STC) builds a spanning tree over the free space and traverses it to guarantee coverage. While it avoids repeated coverage, it may not perform well in narrow or highly constrained spaces such as aircraft aisles [27]. Other graph-based approaches represent the environment using visibility graphs or Voronoi diagrams, treating the CPP task as a graph traversal problem. These are efficient for certain types of structured layouts but are often limited by assumptions about robot geometry or maneuverability.

Beyond classical methods, hybrid strategies have emerged that combine offline planning with reactive behaviors. For instance, Khanam et al. [12] proposed a two-stage CPP framework designed for nuclear facility inspection, where an initial offline map guides the robot through the primary environment, and online adjustments enable coverage in uncertain or hazardous zones. Although developed for a different application domain, the modular structure of this framework—combining macro- and micro-level planning—is relevant to our proposed approach involving reconfigurable cleaning robots in aircraft cabins.

When modeled as a discrete optimization problem, CPP can be reduced to variants of the Traveling Salesman Problem (TSP) or Set Covering Problem (SCP). This abstraction allows the use of powerful metaheuristics to determine optimal or near-optimal visit sequences. A wide range of optimization algorithms have been explored for this purpose. Ant Colony Optimization (ACO) mimics the behavior of ants in finding paths through pheromone-based reinforcement. While effective for small and medium problem sizes, ACO often suffers from slow convergence and sensitivity to parameter tuning [28]. Particle Swarm Optimization (PSO) has been employed in continuous coverage domains, such as drone-based surveillance; however, it must be heavily adapted for discrete path planning and lacks direct applicability in grid-based scenarios. Simulated Annealing (SA), a stochastic method that accepts worse solutions with a probability that decreases over time, has been used in some CPP settings but demands careful control of its cooling schedule [29]. More recently, deep reinforcement learning (RL) approaches have been explored for learning coverage policies from interaction with simulated environments [30], but such methods require large training datasets and long convergence times, which may limit their usability in practical offline planning applications.

Among these techniques, Genetic Algorithms (GAs) have gained significant attention due to their flexibility, simplicity, and proven effectiveness in solving discrete optimization problems such as the Traveling Salesman Problem (TSP) and its variants [31–33]. GAs naturally encode path sequences as chromosomes, making them highly suitable for ordering and optimization tasks in Coverage Path Planning (CPP) [34,35]. Their evolutionary operators—particularly crossover and mutation—allow a balanced exploration and exploitation of the solution space, preventing premature convergence and

improving global optimality [36,37]. Furthermore, GAs are robust to problem-specific constraints, such as avoiding no-go zones, minimizing turning angles, and adapting to dynamic cost functions—features critical for confined, obstacle-rich environments like aircraft interiors [38,39]. Compared with Ant Colony Optimization (ACO) and Particle Swarm Optimization (PSO), which require complex parameter tuning, GAs offer a more straightforward configuration and superior scalability [40]. Moreover, unlike Reinforcement Learning (RL)-based methods, GAs do not rely on large-scale training data and can be directly deployed on preprocessed maps, making them more practical for offline mission planning in real-world robotic cleaning and inspection tasks.

In our proposed two-stage coverage framework, a reconfigurable cleaning robot operates in an aircraft cabin to maximize area coverage. During the first stage, a predefined sweeping pattern ensures primary coverage. However, due to size limitations and geometric constraints, some narrow regions may remain unvisited. At this point, the robot reconfigures into a smaller form, and a GA-based optimizer is triggered to compute the most efficient sequence to cover the remaining uncovered regions. This formulation, which reduces the problem to a partial CPP task resembling TSP with a limited set of goals, plays to the strengths of the Genetic Algorithm in terms of convergence speed, adaptability, and path quality.

## 2.1 Aircraft environment

While many adaptive robotic mapping platforms—such as the Smorphi —demonstrate robust performance in domestic settings using onboard LiDAR and vision sensors, the aircraft cabin introduces a uniquely structured and constrained environment. Unlike residential spaces, aircraft interiors demand a tailored modeling approach due to their strict geometry, high object density, and safety-driven design constraints.

To ensure realism and applicability, our simulation environments are constructed based on actual aircraft specifications sourced from official manufacturing and operations manuals provided by Airbus [41] and Boeing [42]. This study focuses on two representative aircraft models: the Boeing 777−300ER operated by Singapore Airlines and the Airbus A321 operated by Air Seoul. These models were selected due to their operational diversity and representative cabin layouts, providing a rigorous testbed for evaluating the scalability and adaptability of the proposed CPP framework.

the Boeing 777−300ER and Airbus A321-200 are taken as case studies. The airaft cabins present unique challenges for autonomous coverage, including narrow aisles, tight turning radii, and occlusions due to under-seat spaces. Addressing these challenges is critical for demonstrating the robustness of the CPP approach in high-complexity environments.

**2.1.1 Boeing 777−300ER.** For detailed evaluation, the economy class section of the Singapore Airlines Boeing 777−300ER is isolated, which features a standard seat pitch of approximately 32 inches. This section occupies over 50% of the total cabin floor area and contains the highest passenger density, making it an ideal candidate for performance benchmarking.

The reconstructed map and working area dimensions are based on the Boeing 777−300ER operations manual [42] and aligned with the actual seat layout of Singapore Airlines. Fig 3 illustrates the spatial distribution across cabin classes, while Table 1 provides detailed breakdowns of the coverage zones. The economy section alone accounts for an estimated surface area of 140.1 m² targeted for robotic cleaning operations.

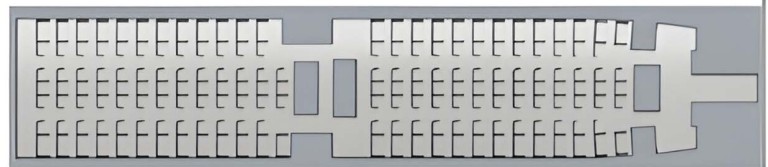

**Fig 3. Working area derivation for the Boeing 777−300ER cabin interior by reconstructed map based on simulated surface scanning.**

**Table 1. Cabin section breakdown for Boeing 777−300ER Air Seoul and Singapore Airlines [42, 43].**

| | Seat Pitch | | Space between seats | | Leg Space | |
|---|---|---|---|---|---|---|
| | inch | mm | inch | mm | inch | mm |
| **Air Seoul cabin configuration** | 29 | 736.6 | 8.1 | 205.7 | 8.2 | 209.2 |
| **Singapore Airlines cabin configuration** | 32 | 812.8 | 10.1 | 256.5 | 12.2 | 310.7 |

To estimate the total cleaning workload across the aircraft, we consider the proportional seat count and layout characteristics of each cabin class. Based on seat count, the ratio of First and Business Class to Economy Class is in Eq 1:

$$\text{Seat Ratio (First + Business) : Economy} = 50 : 228 \tag{1}$$

This implies the economy class constitutes approximately 78.1% of all seats. However, due to the lower seat density and greater surface area per seat in the premium cabins, we apply a conservative adjustment factor of 5% to account for increased cleaning effort. This yields the following relative workload estimates as in Eq 2 and Eq 3:

$$\text{Economy Class Section} = 100\%(\text{baseline}) \tag{2}$$

$$\text{First + Business Class Section} = \text{Additional 26\% (adjusted)} \tag{3}$$

**2.1.2 Airbus A321-200.** In contrast to the 777−300ER, the Airbus A321 operated by Air Seoul features an all-economy layout with two sub-classes differentiated by seat pitch, ranging from 29" to 31" [44] as shown in Fig 4. This layout results in a more uniform coverage area but still presents challenges such as limited maneuvering space between tightly spaced rows.

The Airbus A321-200 cabin layout is benchmarked to be a scanned surface area of 112.5 m². Though smaller than the Boeing 777−300ER, the relatively uniform configuration allows for comparative evaluation of cleaning coverage efficiency and path optimization strategies.

## 3 Reconfiguration platform

To validate the CPP framework, a reconfigurable Footprint-Adaptive Omnidirectional Robot (FAOR) maned Smorphi is employed as in Fig 5. This platform integrates holonomic mobility with adaptive morphology, enabling the robot to dynamically alter its footprint geometry according to environmental constraints. Such capability allows the robot to navigate efficiently across heterogeneous aircraft cabin regions—ranging from wide aisles to confined under-seat areas. Within the proposed CPP formulation, the FAOR's kinematic, energetic, and morphological characteristics are embedded as optimization variables, allowing configuration decisions to be co-optimized with the global coverage path sequence.

### 3.1 Representation of the reconfigurable robot in a workspace

The workspace representation of the reconfigurable robot extends the configuration-space ($\mathcal{C}$-space) modeling approach introduced in [45]. The configuration vector of the four-block modular robot is defined in Eq 4 as $\mathbf{q} \in \mathcal{C} = \mathbb{R}^2 \times \mathbb{S}^1 \times \mathbb{S}^4$,

$$\mathbf{q} = [x, y, \theta, \phi_1, \phi_2, \phi_3, \phi_4]. \tag{4}$$

Here, $[x, y]$ denotes the robot's Cartesian position, $\theta \in \mathbb{S}^1$ represents its body orientation, and $\phi_1, \phi_2, \phi_3, \phi_4 \in \mathbb{S}^1$ correspond to the hinge joint angles of the modular segments relative to a reference axis.

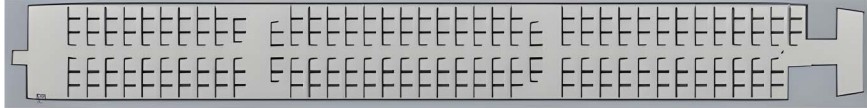

**Fig 4. Working area derivation for the Airbus A321-200 cabin interior by simulated map from surface scanning.**

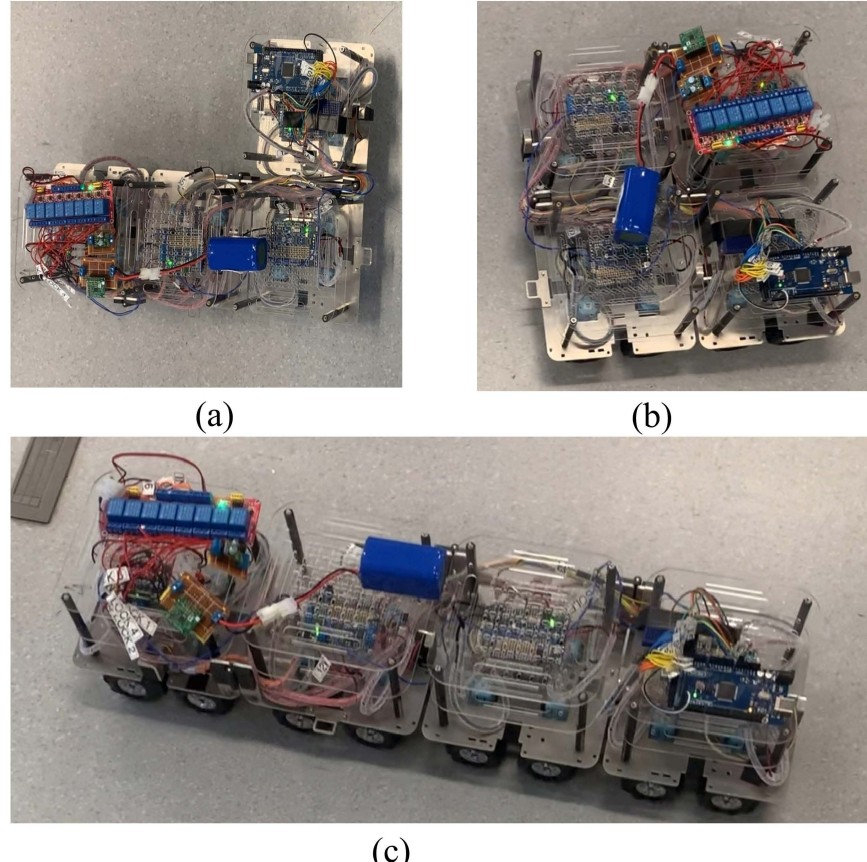

(a)   (b)

(c)

**Fig 5. The Smorphi reconfigurable robot with modular design, dimensioned for aircraft cabin environments: A shape-shifting transition between configurations.** B: full-size configuration ($C_L$) covering 0.096 m². C: half-size configuration ($C_S$) covering 0.024 m².

Three elementary action types are defined: (1) *translation*, representing planar motion of the entire structure; (2) *rotation*, denoting global orientation change; and (3) *transformation*, referring to actuation of the hinge joints to achieve compression or expansion. A reconfiguration step is thus represented as a transition between two configurations, $\mathbf{q}_I$ and $\mathbf{q}_D$, quantified by the total angular displacement of the hinges as in Eq 5:

$$\Delta\phi_b = |\phi_b^{(D)} - \phi_b^{(I)}|, \quad b \in \{1, 2, 3, 4\}. \tag{5}$$

This displacement contributes to the transformation cost, ensuring that morphological adaptation is directly embedded within the total energy model. Table 2 summarizes an example of hinge rotation magnitudes for transitions between compressed and expanded configurations.

### 3.2 Configuration modes and energy modeling

The simulated Smorphi robot is designed to emulate the physical FAOR platform, comprising four modular segments capable of dynamically reconfiguring to meet varying spatial constraints within an aircraft cabin. Two primary operational configurations are defined: the full-size configuration ($C_L$) and the half-size configuration ($C_S$). Each mode presents unique trade-offs in terms of footprint coverage, maneuverability, and energy consumption as shown in Fig 5.

The robot's dimensions were determined with reference to standard Airbus A321 and Boeing 777 cabin geometries to ensure operational compatibility. In its full-size configuration ($C_L$), the robot covers approximately 0.096 m² (0.32 m × 0.30 m), suitable for open regions such as aisles and galley areas where clearances exceed 0.50 m. The associated motion energy consumption is modeled asin Eq 6:

$$W(C_L) = W_{max}, \quad E_{move}(C_L, d) = \alpha \cdot d,$$
(6)

where $W(C_L)$ denotes the effective operational width, $d$ is the traveled distance, and $\alpha$ represents the motion energy coefficient (Wh/m).

In contrast, the half-size configuration ($C_S$) covers approximately 0.024 m² (0.16 m × 0.15 m), allowing it to access restricted areas such as under-seat spaces and sidewall gaps. Although it enhances reachability, this mode entails higher energy expenditure due to increased actuation and reduced traction efficiency as in Eq 7:

$$W(C_S) = W_{min}, \quad E_{move}(C_S, d) = \beta \cdot d, \quad \beta > \alpha.$$
(7)

Here, the inequality $\beta > \alpha$ captures the relative inefficiency of locomotion in the compact mode.

Within the proposed hybrid CPP framework, the configuration mode ($C_L$ or $C_S$) is jointly optimized with the waypoint visiting sequence during the GA-based adaptive TSP optimization in Stage 2. The Genetic Algorithm simultaneously determines both the optimal path and the most suitable configuration for each segment, thereby achieving a balance between coverage efficiency, energy minimization, and spatial adaptability across the aircraft cabin environment.

### 3.3 Data preparation – aircraft cabin components

To ensure realistic representation of the operational workspace, detailed spatial data describing seat dimensions, aisle widths, and service zone geometries were extracted from manufacturer specifications and airline layout plans. Table 3 compares the seating configurations of the two aircraft studied—Singapore Airlines' Boeing 777−300ER and Air Seoul's Airbus A321-200. The A321-200, used as a stress-test case, features a denser cabin arrangement with seat pitches as small as 29 inches, representing a worst-case operational scenario for robotic maneuverability.

**Table 2. Angular displacements $\Delta\phi_b$ for transformation between configuration states (in radians).**

| $q_iq_D$ | Compressed State | Expanded State |
|---|---|---|
| Fully Compressed | [0, 0, 0, 0] | $[\frac{\pi}{2}, 0, \frac{\pi}{2}, 0]$ |
| Fully Expanded | $[-\frac{\pi}{2}, 0, -\frac{\pi}{2}, 0]$ | [0, 0, 0, 0] |

**Table 3. Pitch and cabin configuration of both aircrafts.**

|  | Seat Pitch | | Space between seats | | Leg Space | |
|---|---|---|---|---|---|---|
|  | inch | mm | inch | mm | inch | mm |
| **Air Seoul** | 29 | 736.6 | 8.1 | 205.7 | 8.2 | 209.2 |
| **Singapore Airlines** | 32 | 812.8 | 10.1 | 256.5 | 12.2 | 310.7 |

Unlike human cleaning crews, who can flexibly adjust posture to reach under seats, a cleaning robot is mechanically constrained by its physical envelope. Measurements taken from the A321-200 cabin indicate an average vertical clearance of 8.1 inches between seat rows, expanding slightly to 8.2 inches at floor level. These parameters establish a critical geometric threshold for autonomous cleaning operations. As illustrated in Fig 6, the robot must navigate through narrow gaps and around structural obstructions to reach debris accumulation zones beneath passenger seats.

The reconfigurable design of the Smorphi robot is specifically designed to address the spatial variability within aircraft cabins. While both the full-size and half-size configurations are capable of accessing all cabin areas, their operational efficiency differs depending on local geometric constraints. For instance, in densely packed regions such as inter-seat spaces within economy-class sections, the half-size configuration (CS) enables smoother traversal and reduced collision risk. Although the full-size configuration (CL) can also reach these areas, it experiences limited maneuverability due to its larger footprint. Consequently, CL is preferentially employed in more open regions—such as aisles, galleys, and entryways—where it provides greater stability and energy efficiency. This adaptive switching mechanism allows Smorphi to maintain uniform cleaning performance throughout the cabin, even under severe spatial restrictions. The detail CPP framework is described in detail in the following section.

## 4 Proposed complete coverage path planning framework for aircraft cabin cleaning

Aircraft cabin cleaning presents distinct challenges arising from the constrained and structured layout defined by the Layout of Passenger Accommodation (LOPA), as well as the presence of unpredictable obstacles such as trash, personal belongings, and varying seat recline positions. The proposed Coverage Path Planning (CPP) framework addresses these challenges by integrating deterministic sweeping patterns for general area coverage with an adaptive optimization process for waypoint sequencing. In the present study, the cleaning task is modeled under the assumption of a vacuum-based

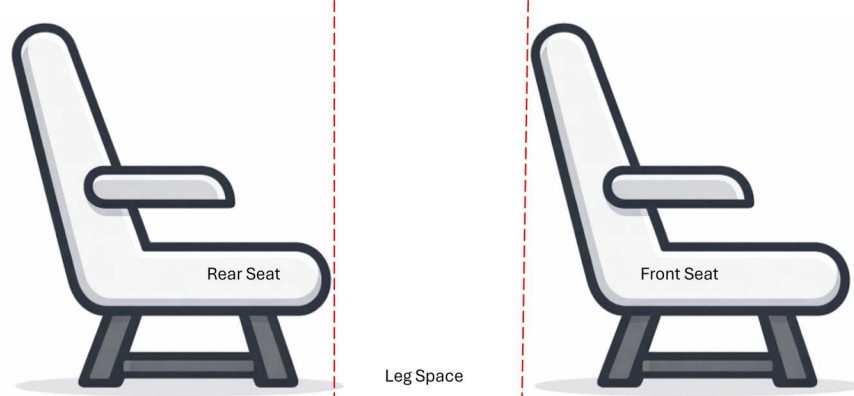

**Fig 6. Clearance conditions for cleaning between aircraft seats.**

mechanism, and sterilization-specific constraints such as UV exposure dosage or spray dispersion are not explicitly considered.

### 4.1 Overview of the two-stage CPP framework

As shown in Fig 7, the robotic cleaning process is designed to operate under real-world aircraft turnaround constraints. The aircraft is assumed to have predictable layouts (e.g., Boeing 777−300ER and Airbus A321-200), but often includes obstacles such as blankets, large food packaging, or personal belongings left behind. These factors require adaptive response beyond preplanned paths.

After process A for cabin attendant disembarking and before process D for cleaner disembarking as depicted in the Fig 8, the automated cleaning framework operates in two stages:

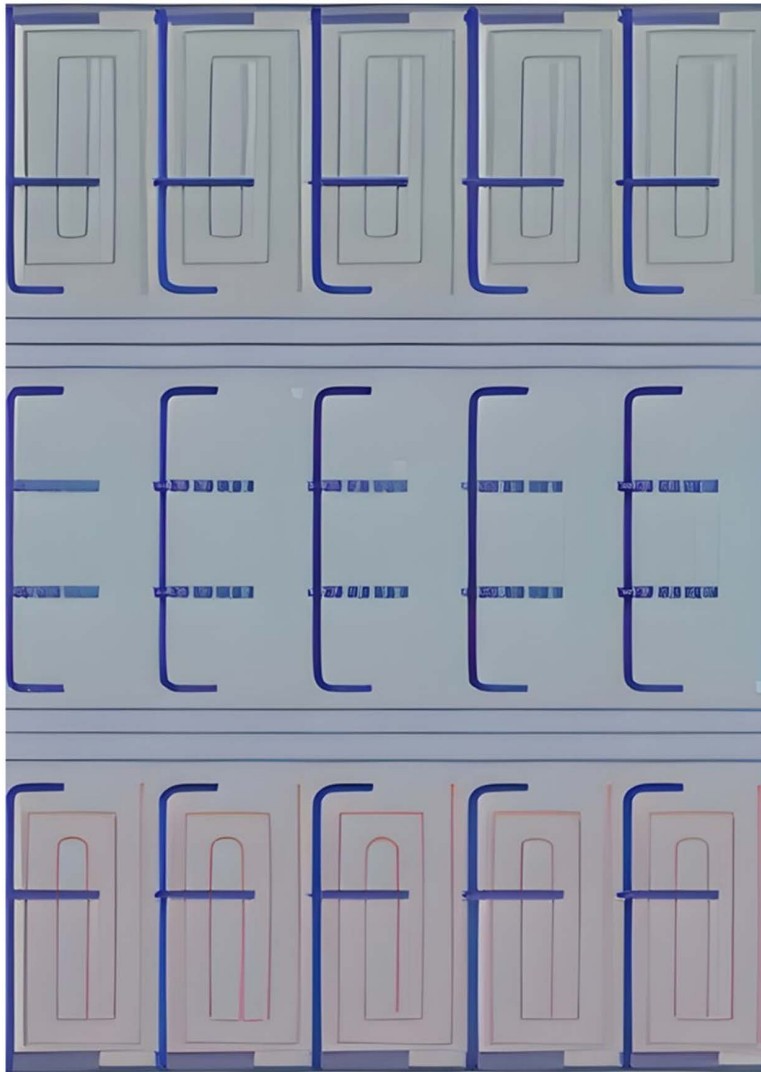

**Fig 7. Model depicting seat structure anatomy and path navigation of robot within the cabin.**

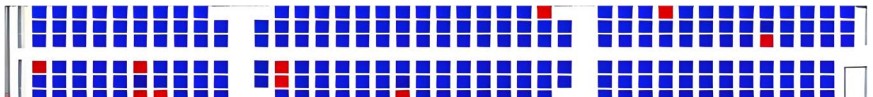

**Fig 8. CPP Framework showing integration of scanning, initial avoidance, and secondary local cleaning.**

- **Stage 1: Overall Coverage (Process B)** – The full-size robot performs general sweeping based on a predefined zigzag or spiral pattern, skipping over detected obstacles and blocked rows (Fig 9). In Stage 1, seat rows are classified as hard-to-access waypoints when the available free-space clearance, derived from the reconstructed cabin layout, falls below the maneuverability envelope of the robot in its full configuration or when geometric obstructions prevent safe traversal. This determination is performed at the map and simulation level using known aircraft cabin geometry and robot kinematic constraints. The current study focuses on planning-level feasibility, and that perception uncertainty and sensor noise are outside the scope of this work.

- **Stage 2: Adaptive Local Coverage (Process C)** – A half-size robot is deployed to revisit waypoints identified in Stage 1 as skipped or obstructed. These waypoints visting order are optimized using a Genetic Algorithm (GA)-based planner, solving a partial-CPP problem that resembles the Traveling Salesman Problem (TSP) guaranteeing deterministic revisitation for CPP in Stage 2. Accordingly, the term CPP refers to the proposed two-stage framework as a whole, rather than to the partial sweeping behavior of Stage 1 in isolation

The contributions of each cleaning strategy and corresponding robot size in the aircraft turnaround operation are summarized in Table 4.

### 4.2 Stage 1: Structured coverage with spiral and Zigzag patterns

In Stage 1, the framework focuses on structured and complete coverage of the aircraft cabin through systematic path planning patterns, namely the Spiral and Zigzag patterns. Stage 1 is intentionally designed as a rapid and conservative sweeping process under the robot's full configuration, and therefore does not aim to achieve complete coverage on its own. Regions that are not covered due to accessibility constraints or conservative decisions are explicitly recorded as waypoints and deferred to Stage 2 for guaranteed revisitation.

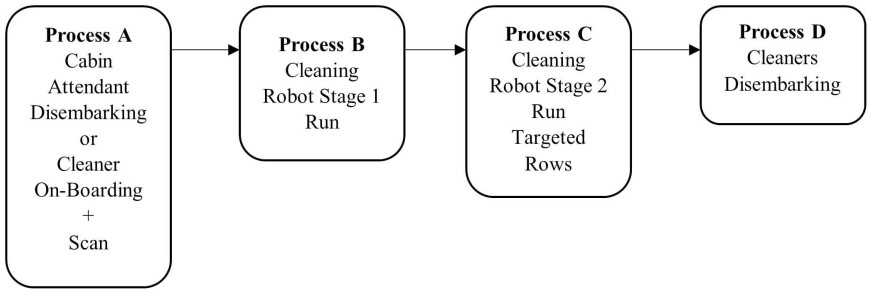

**Fig 9. Example of Process A–B: Full-size robot skips obstructed rows due to unremoved items.**

**Table 4. Summary of roles for each process in the CPP framework.**

| Stages | Robot Operation | Cabin Attendant/Cleaner Operation |
|---|---|---|
| **Process A** | | |
| *Cabin Attendant Disembarking & Cleaning Crew n-Boarding* | • Cleaning Robot will be moved to starting points, within the cabin<br>• No operations on the cleaning robots | • Cabin Attendants, as one of disembarking procedure, about to leave the aircraft Cleaning<br>• Crew, as one of the boarding procedures, about to board aircraft to begin cleaning<br>• To mark out rows of seats with obstacles |
| **Process B** | | |
| *1ˢᵗ Run – Entire Cabin* | • Robot will take note of the marked-out areas from the scan<br>• Robot will begin run from home position<br>• Adopt cleaning pattern throughout aircraft cabin<br>• And skip these marked rows entirely<br>• Robot will finish run of the entire cabin | • Cleaner will take this chance to visit marked rows<br>• And physically remove rubbish<br>• Other cleaners use this opportunity to clean other cabin monuments |
| *2ⁿᵈ Run – Targeted Rows* | • Robot will begin from end point of the aircraft<br>• With all skipped areas of the cabin<br>• Revisit all skipped areas of the cabin | • No Operation |
| **Process D** | | |
| *Cleaners Disembarking* | • No Operation | • Cleaners will disembark<br>• Aircraft should be cleaned for subsequent flight |

**Spiral pattern cleaning.** The spiral cleaning pattern, commonly used for open spaces, typically involves a circular trajectory where the robot spirals inward or outward along a rotational path [46]. However, given the rectangular and highly structured layout of an aircraft cabin, our study adopts a square spiral pattern. This adaptation ensures consistent and thorough cleaning along the linear edges and corners of the subregions.

In this approach, the aircraft cabin is partitioned into multiple subregions or cells. Within each cell, the robot follows a square-spiral trajectory composed of straight segments and 90° turns. This design guarantees complete surface coverage in confined and rectilinear spaces. If the robot encounters any deviation or minor obstacle, it performs corrective actions and resumes the spiral motion from the appropriate segment, continuing until the entire cell is covered. It then proceeds to the next cell, initiating a new square spiral pattern. The full behavior of this spiral-patterned coverage is illustrated in Fig 10.

**Zigzag Pattern Cleaning.** In addition to the spiral method, Stage 1 also incorporates a zigzag (or boustrophedon) pattern [47,48], which is especially effective in long and narrow spaces such as aircraft aisles or seat rows. This method involves straight-line motion along the width or length of the cabin, with 180° turns at the boundaries to reverse direction and begin a new sweep.

As with the spiral strategy, the cabin is divided into discrete rectangular cells. The robot visits each cell in sequence and performs a back-and-forth zigzag motion within each. Upon encountering an obstacle during its traversal, the robot executes a predefined maneuver sequence to bypass the obstruction:

• Move backward

• Turn 90° (either left or right)

• Move forward

• Turn 90° (opposite direction)

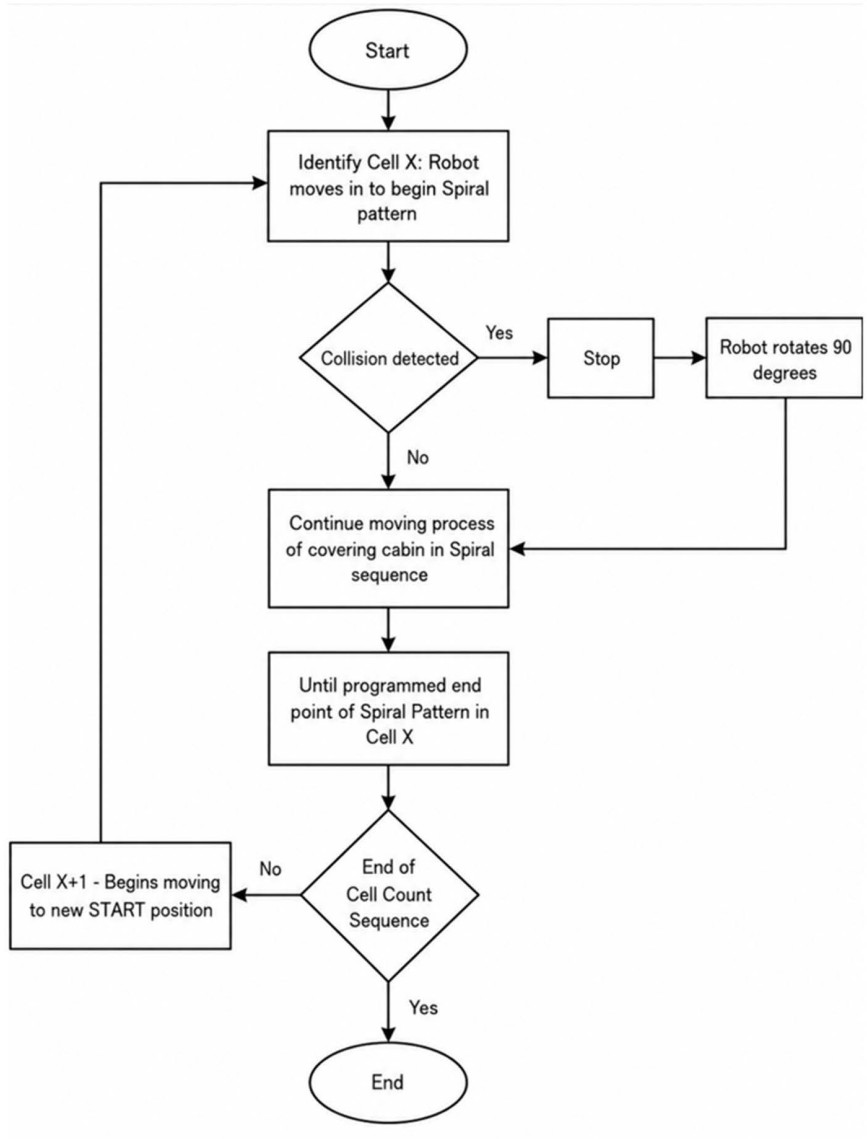

**Fig 10. Spiral-patterned CPP approach for navigating the aircraft cabin interior.**

Once clear of the obstacle, the robot resumes the zigzag coverage pattern within the same cell before proceeding to the next. This recovery strategy ensures minimal disruption to the overall path plan. The complete zigzag-patterned coverage behavior is depicted in Fig 11.

The combination of square-spiral and zigzag patterns allows the robot to adapt its motion to different sections of the aircraft—handling seat rows, galleys, and aisles efficiently—while ensuring complete and non-redundant coverage.

The proposed two-stage CPP framework allows real-time responsiveness in a semi-structured aircraft cabin. The GA-based adaptive planning in Stage 2 enables robust and efficient cleaning in rows initially skipped due to physical obstacles. This hybrid framework achieves high coverage rates, minimizes idle travel, and aligns with the dynamic nature of real-world aircraft cleaning scenarios.

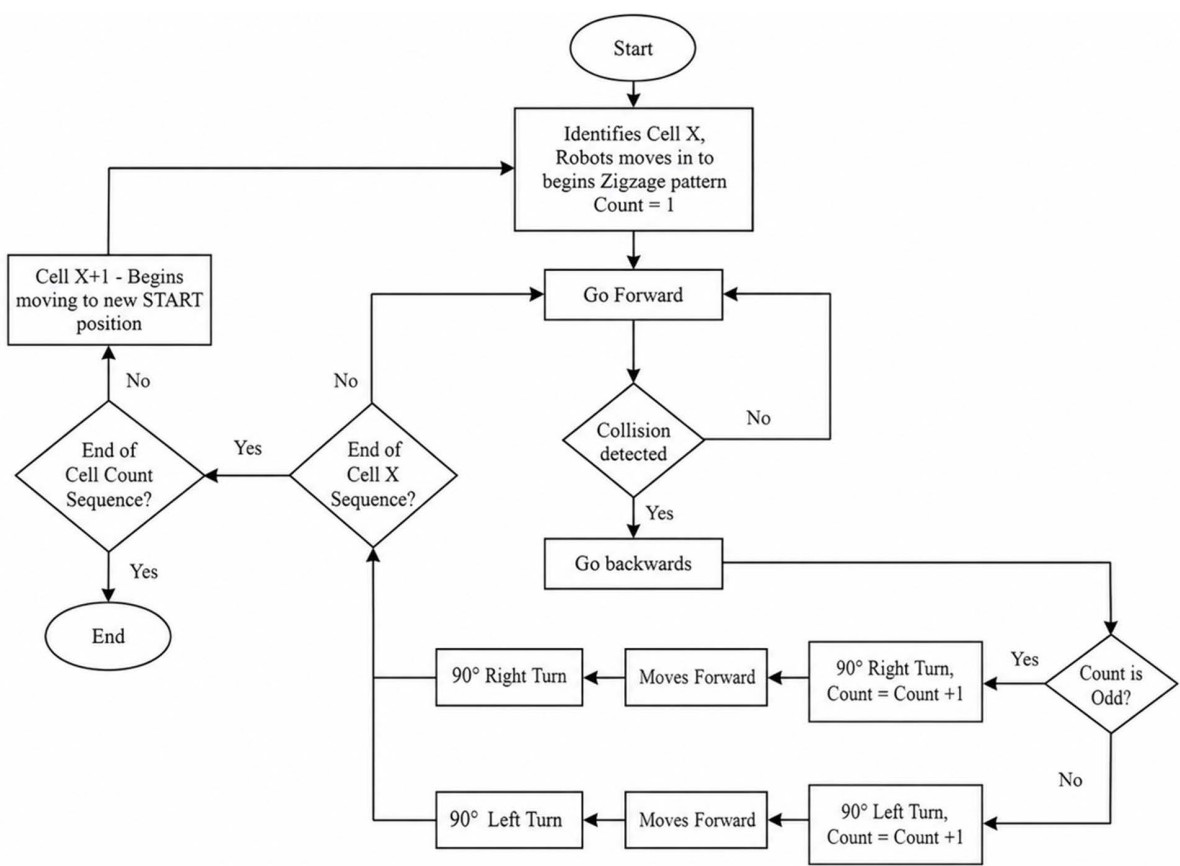

**Fig 11. Zigzag-patterned CPP approach for navigating the aircraft cabin interior.**

**Algorithm 1 GA-Based Adaptive Path Optimization for Smorphi in Stage 2 Cleaning ($W$, $C$, $P$, $G$)**

```
1.  Pop ← InitializePopulation(W, C, P)
2.  Π* ← ∅
3.  T_C* ← ∞
4.  for g = 1, ..., G do
5.      forEach Π = [W', c] ∈ Pop do
6.          T_C ← ComputeTotalCost(Π)
7.          f(Π) ← 1/T_C
8.      Parents ← TournamentSelection(Pop)
9.      Offspring ← ∅
10.     forEach pair (p₁, p₂) ∈ Parents do
11.         Π_child ← EdgeRecombinationCrossover(p₁, p₂)
12.         if U([0, 1]) < p_m then
13.             Π_child ← WaypointSwapMutation(Π_child)
14.         if U([0, 1]) < p_r then
15.             Π_child ← ConfigToggleMutation(Π_child, C_L, C_S)
16.         Offspring ← Offspring ∪ {Π_child}
17.     Pop ← SurvivorSelection(Pop, Offspring)
18.     Π_best ← argmin(T_C)
                   Π∈Pop
19.     if T_C(Π_best) < T_C* then
20.         Π* ← Π_best
21.         T_C* ← T_C(Π_best)
22. return Π*: optimal cleaning path and configuration sequence.
```

### 4.3 Genetic algorithm-based reconfigurable path optimization for process C (stage 2)

After completing the Zig-zag coverage pattern in Stage 1 of Process B (Fig 8), certain regions—such as under-seat areas and narrow aisle intersections—may remain partially uncleaned due to the spatial constraints of the full-size Smorphi configuration. To address these residual regions, Process C activates Stage 2 cleaning, during which the Smorphi robot autonomously reconfigures to an appropriate size mode and performs localized cleaning around the identified problematic areas.

Inspired by the classical Traveling Salesman Problem (TSP) [49], Stage 2 is formulated as an optimization problem that determines an efficient visiting sequence among dynamically generated cleaning waypoints. In the proposed framework, the waypoint set $W$ is defined at the seat level rather than at the row level. Multiple waypoints may therefore be generated within the same seat row, with each waypoint corresponding to an individual seat or under-seat coverage region that was identified as skipped during Stage 1. Each waypoint is positioned at the centroid of its associated local coverage area, establishing a one-to-one mapping between waypoints and the physical regions requiring additional cleaning. This formulation enables fine-grained coverage control, as only the specific seats that remain uncovered after Stage 1 are revisited during Stage 2, rather than enforcing a coarse row-level revisitation strategy. The objective of Stage 2 in Process C is to minimize the total travel distance and morphological reconfiguration cost while ensuring complete coverage by optimally visiting all defined waypoints.

The optimization framework is implemented using a Genetic Algorithm (GA), following the principles of [50], but extended to incorporate Smorphi's reconfigurable behavior. The GA algorithm, as detailed in Algorithm 1 operates over a set of seat-row waypoints $W = \{w_1, w_2, \ldots, w_n\}$, representing the target cleaning locations, and a set of configuration modes $C = \{C_L, C_S\}$, corresponding to the full-size and half-size operational states of Smorphi.

The population size $P$ defines the number of candidate solutions (chromosomes) evaluated per generation, while the number of generations $G$ determines the total iterations for evolutionary refinement. Each chromosome encodes both a waypoint permutation and a configuration sequence $\mathbf{c} = [c_1, c_2, \ldots, c_n]$, where $c_i \in C$ specifies the robot configuration used to reach waypoint $w_i$.

In each generation, the total cost $T_C$ of every candidate path as in Eq 8 is computed, incorporating both travel distance and the energy expenditure from configuration transitions. The fitness function is defined as $f(\Pi) = 1/T_C$, rewarding shorter and more energy-efficient trajectories. Tournament selection identifies high-performing individuals to form the parent set for crossover. Edge Recombination Crossover (ERX) is employed to preserve waypoint adjacency relationships, maintaining spatial continuity between sequential targets.

$$T_C = \sum_{i=1}^{n-1} \left( \gamma_1 \cdot D(w_i, w_{i+1}) + \gamma_2 \cdot E_{\text{move}}(c_i, D(w_i, w_{i+1})) + \gamma_3 \cdot E_{\text{reconf}}(c_i, c_{i+1}) \right)$$

(8)

where $D(w_i, w_{i+1})$ denotes the Euclidean distance between consecutive waypoints, $E_{\text{move}}$ represents the energy required for motion under configuration $c_i$, and $E_{\text{reconf}}$ quantifies the energy/time cost of transitioning between configurations. In this study, the motion energy term $E_{\text{move}}$ is modeled as a linear function of travel distance to maintain computational efficiency and enable fair comparison among different optimization algorithms.

The weighting coefficients $\gamma_1$, $\gamma_2$, and $\gamma_3$ are empirically tuned through preliminary trials to balance travel distance, motion-related energy consumption, and morphology reconfiguration cost, respectively. The selected values are chosen to provide stable convergence and reasonable trade-offs under typical aircraft cabin layouts. Although increasing $\gamma_3$ biases the optimization toward solutions with fewer morphology reconfigurations, coverage completeness is not affected, as all waypoints must be visited exactly once by construction of the waypoint-based formulation.

Mutation operators introduce diversity in two forms: (i) a waypoint swap mutation, which exchanges two waypoints to explore alternative route orderings, and (ii) a configuration toggle mutation, which switches the configuration mode between $C_L$ and $C_S$ to adapt morphology to spatial constraints. The survivor selection step replaces the least-fit individuals with offspring, ensuring evolutionary pressure toward improved solutions. The algorithm iteratively refines the population until convergence or the maximum generation $G$ is reached. The final output $\Pi^*$ represents the optimal path and corresponding configuration sequence that minimizes total cleaning cost $T_C^*$, effectively balancing spatial adaptability, energy efficiency, and travel time within the aircraft cabin environment.

As a result, Stage 2 cleaning is executed efficiently with minimal redundant motion and adaptive structural changes tailored to cabin constraints. The adaptive GA planner ensures energy- and time-efficient traversal without full mission replanning, enhancing robustness to environmental variations, supporting selective re-cleaning, and complementing Stage 1 to form a unified two-stage CPP framework for confined aircraft cabin environments.

### 4.4 Experimental setup

To validate the proposed Coverage Path Planning (CPP) framework, a series of experiments were conducted using the reconfigurable Smorphi robot within two simulated aircraft cabin environments. The experimental setup was designed to evaluate both stages of the cleaning framework: (1) global coverage using deterministic patterns and (2) adaptive local re-cleaning optimized through a Genetic Algorithm (GA)-based Traveling Salesman Problem (TSP) formulation. The Genetic Algorithm (GA) parameters were determined empirically through a series of experimental trials to achieve optimal performance for Smorphi's autonomous cleaning task. The final configuration of parameters is summarized as follows:

- Population size ($P$): 20 candidate paths are maintained per generation to balance computational cost and convergence.

- Number of generations ($G$): Set empirically based on the number of target waypoints (e.g., $G = 100$ for 50 waypoints).

- Crossover operator: Edge Recombination Crossover (ERX) preserves adjacency between waypoints, reflecting realistic movement continuity.

- Mutation operator: Dual-mode mutation is applied — waypoint inversion (probability $p_m = 0.4$) and configuration toggling ($p_r = 0.2$) — to maintain diversity and discover efficient configuration-path pairings.

- Fitness function: Each chromosome is evaluated as $f(\Pi) = 1/T_C$, promoting shorter and more energy-efficient paths.

- Selection strategy: Tournament selection ensures that high-performing individuals dominate evolution while retaining stochastic diversity.

**4.4.1 Simulation environment.** The test environments were modeled based on the cabin layouts of the Singapore Airlines Boeing 777−300ER and the Air Seoul Airbus A321-200. Both models were reconstructed from publicly available aircraft seat maps and structural blueprints, including seat pitch, aisle width, and under-seat clearance dimensions. These datasets were integrated into a custom simulation environment developed in Python and ROS environment, where the robot's motion kinematics, collision avoidance, and coverage computation were implemented.

Each environment includes static obstacles representing seats, walls, and service partitions. The simulation environment models macro-level cabin geometry derived from aircraft layout diagrams, including seat arrangement, aisle width, and under-seat clearances. Micro-scale surface variations such as carpet texture, floor thresholds, and metal seat rails are not explicitly represented. The cleaning zones were segmented into two operational layers: (General Coverage): the full-size Smorphi configuration executes a Zigzag or Spiral pattern to cover the cabin floor uniformly. (Targeted Re-cleaning): The half-size configuration is deployed to navigate restricted areas such as under-seat zones or narrow aisles. Waypoints for this phase are automatically generated from areas skipped or marked as low-confidence coverage in Stage 1.

The Smorphi robot is modeled as a four-module reconfigurable platform capable of switching between two primary shapes: a full-size mode for wide-area cleaning and a half-size mode for accessing confined spaces. The transformation between configurations incurs an energy cost proportional to the hinge displacement, as formulated in Section 3. During simulation, this reconfiguration energy is incorporated into the overall travel cost

## 4.5  Results for process B using spiral and Zigzag coverage algorithms

To establish a performance baseline, reference data from domestic home environment experiments were used to define benchmark expectations for coverage and mobility performance. Specifically, the expected coverage percentage ranged between 87.7% and 93%, while the average cleaning speed was benchmarked between 0.9 m²/min and 1.1 m²/min, with an allowable deviation of ±0.1 m²/min. These benchmarks serve as comparative references for evaluating the simulated cleaning performance within the Airbus A321-200 and Boeing 777−300ER cabin environments.

Following the completion of experiments for Process B within the proposed CPP framework, the Spiral coverage pattern results with 50 seats skipped for Process C are summarized in Table 5, with corresponding trajectory visualizations presented in Fig 12. Similarly, the Zigzag coverage pattern results for 50 seats senarios are shown in Table 6, with the associated trajectory illustrated in Fig 13.

**Table 5.  Performance of Spiral Coverage Path on Boeing 777−300ER and Airbus A321 with 50 seats skipped for Process C.**

| Trials | Path Length (m) | Traveling Time (s) | Total Area Covered (m²) | Coverage (%) | Average Moving Speed (m²/min) |
|---|---|---|---|---|---|
| Boeing 777−300ER (Spiral) – 140.1 m² | | | | | |
| Trial 1 | 884.9 | 7493 | 121.4 | 86.7 | 1.03 |
| Trial 2 | 884.9 | 7947 | 120.6 | 86.1 | 0.97 |
| Trial 3 | 884.9 | 8012 | 120.1 | 85.7 | 0.95 |
| Trial 4 | 884.9 | 6746 | 122.0 | 87.0 | 1.15 |
| Trial 5 | 884.9 | 6664 | 123.0 | 87.4 | 1.17 |
| Trial 6 | 884.9 | 7283 | 121.3 | 86.6 | 1.06 |
| Trial 7 | 884.9 | 7569 | 120.6 | 86.1 | 1.01 |
| Trial 8 | 884.9 | 8231 | 119.5 | 85.3 | 0.92 |
| Trial 9 | 884.9 | 8222 | 119.6 | 85.4 | 0.92 |
| Trial 10 | 884.9 | 7872 | 120.3 | 85.9 | 0.97 |
| **Mean** | **884.9** | **7613.9** | **120.9** | **86.2** | **1.01** |
| Airbus A321 (Spiral) – 112.5 m² | | | | | |
| Trial 1 | 703.6 | 6626 | 97.0 | 86.2 | 0.92 |
| Trial 2 | 703.6 | 6623 | 96.8 | 86.0 | 0.92 |
| Trial 3 | 703.6 | 5812 | 98.0 | 87.0 | 1.06 |
| Trial 4 | 703.6 | 5808 | 98.2 | 87.2 | 1.06 |
| Trial 5 | 703.6 | 5811 | 97.5 | 86.9 | 1.06 |
| Trial 6 | 703.6 | 5705 | 98.2 | 87.3 | 1.09 |
| Trial 7 | 703.6 | 6391 | 97.0 | 86.2 | 0.96 |
| Trial 8 | 703.6 | 6544 | 96.4 | 85.8 | 0.93 |
| Trial 9 | 703.6 | 6632 | 96.3 | 85.7 | 0.92 |
| Trial 10 | 703.6 | 6316 | 97.1 | 86.3 | 0.97 |
| **Mean** | **703.6** | **6226.8** | **97.2** | **86.3** | **0.99** |

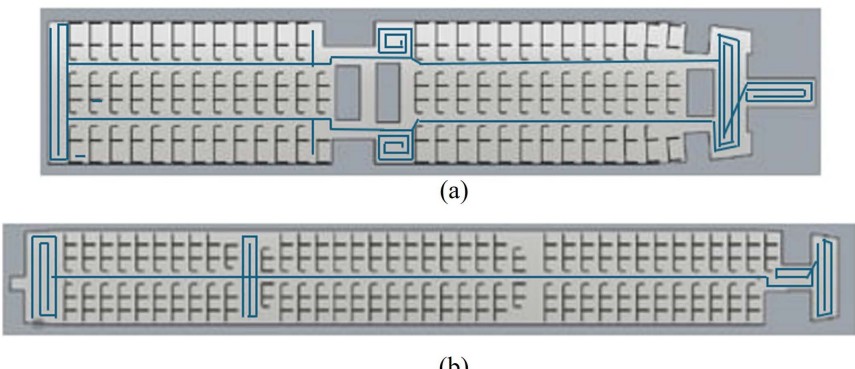

(a)

(b)

**Fig 12. Depicts the Spiral pattern of the cleaning robot through both aircraftby mapping output from the robot travel for (a) Airbus 321−200 (b) Boeing 777−300ER.**

**Table 6. Performance of Zig-zag Coverage Path on Boeing 777−300ER and Airbus A321 with 50 seats skipped for Process C.**

| Trials | Path Length (m) | Traveling Time (s) | Total Area Covered (m²) | Coverage (%) | Average Moving Speed (m²/min) |
|---|---|---|---|---|---|
| Boeing 777−300ER (Zig-zag) – 140.1 m² | | | | | |
| Trial 1 | 831.7 | 7129 | 114.2 | 81.5 | 0.96 |
| Trial 2 | 831.3 | 6597 | 113.8 | 81.2 | 1.04 |
| Trial 3 | 831.7 | 6621 | 114.0 | 81.4 | 1.03 |
| Trial 4 | 831.7 | 7416 | 113.9 | 81.3 | 0.92 |
| Trial 5 | 831.7 | 7078 | 114.3 | 81.6 | 0.97 |
| Trial 6 | 831.7 | 6604 | 113.9 | 81.3 | 1.04 |
| Trial 7 | 831.7 | 7193 | 114.1 | 81.5 | 0.95 |
| Trial 8 | 831.7 | 6766 | 113.8 | 81.2 | 1.01 |
| Trial 9 | 831.7 | 6733 | 114.2 | 81.5 | 1.02 |
| Trial 10 | 831.7 | 6849 | 114.1 | 81.5 | 1.00 |
| **Mean** | **831.7** | **6949** | **114.0** | **81.4** | **0.99** |
| Airbus A321 (Zig-zag) – 112.5 m² | | | | | |
| Trial 1 | 667.7 | 5132 | 91.2 | 81.1 | 1.07 |
| Trial 2 | 667.7 | 5125 | 91.4 | 81.3 | 1.07 |
| Trial 3 | 667.7 | 6054 | 91.1 | 80.9 | 0.90 |
| Trial 4 | 667.7 | 5665 | 91.3 | 81.2 | 0.97 |
| Trial 5 | 667.7 | 5387 | 91.5 | 81.4 | 1.02 |
| Trial 6 | 667.7 | 5968 | 91.1 | 81.0 | 0.92 |
| Trial 7 | 667.7 | 6966 | 91.2 | 81.1 | 0.79 |
| Trial 8 | 667.7 | 5330 | 91.2 | 81.1 | 1.03 |
| Trial 9 | 667.7 | 5876 | 91.3 | 81.2 | 0.93 |
| Trial 10 | 667.7 | 5916 | 91.2 | 81.1 | 0.93 |
| **Mean** | **667.7** | **5742** | **91.2** | **81.2** | **0.96** |

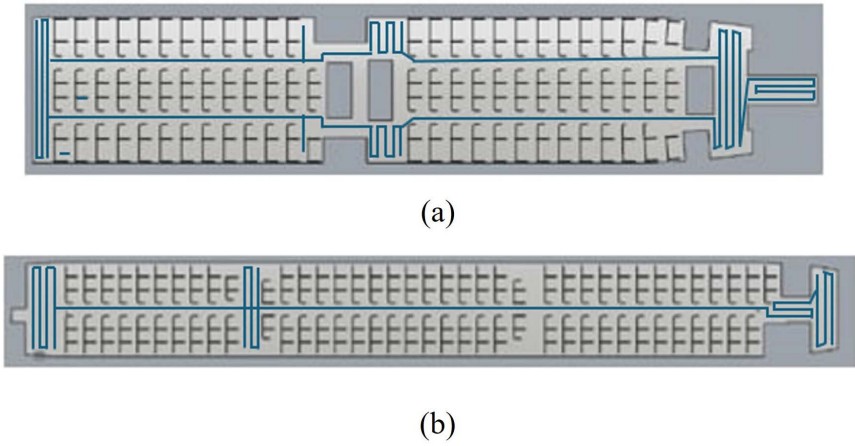

(a)

(b)

**Fig 13. Depicts the Zigzag pattern of the cleaning robot through both aircraft by mapping output from the robot travel for (a) Airbus 321−200 (b) Boeing 777−300ER.**

The experimental results show that the Spiral cleaning pattern achieved coverage rates of 86.3% in the Airbus A321-200 and 86.2% in the Boeing 777−300ER. In comparison, the Zig-zag cleaning pattern yielded slightly lower coverage values of 81.2% and 81.4%, respectively. Although the differences in coverage are moderate, the Zig-zag method consistently enabled higher cleaning speeds and shorter overall cleaning durations, as reported in Tables 5 and 6.

The extended results for the 100-seat scenarios for both spiral and zig-zag are in Tables 7 and 8, respectively. Both spiral and zig-zag patterns exhibit proportional increases in path length and traveling time relative to the 50-seat case, while maintaining stable coverage rates and average moving speeds. This indicates that the coverage behavior scales predictably with task size and does not introduce unexpected inefficiencies as the number of skipped seats increases. Across both aircraft models, the spiral coverage pattern consistently achieves higher coverage ratios (approximately 86–87%) compared to the zig-zag pattern (approximately 81%), at the cost of increased path length and travel time.

To further evaluate cleaning performance, Process B was also tested in both aircraft and home environments. As shown in Fig 14 and Table 9, the coverage achieved inside aircraft cabins was approximately 5–7% lower than that observed in Sample Houses 1 and 2. This reduction is expected, as aircraft cabins constitute highly constrained environments with narrow aisles, dense seating layouts, and numerous geometric obstacles that limit the robot's ability to achieve full coverage.

Overall, these findings validate the feasibility and robustness of the proposed CPP framework in structured yet constrained environments such as aircraft cabins. Although the Spiral pattern offers slightly more uniform coverage, the Zigzag cleaning strategy demonstrates superior operational efficiency, making it the preferred approach for practical deployment. Its ability to balance cleaning completeness with time efficiency aligns well with the strict turnaround time requirements of real-world airline maintenance operations.

### 4.6 Results for process C using GA TSP optimization-based algorithms

This section presents the experimental results for Process C of the proposed CPP framework, where the cleaning robot is required to visit between 10 and 100 randomly distributed seat locations identified as requiring cleaning. This subtask is formulated as a partial Traveling Salesman Problem (TSP), optimized using the proposed GA-based method

**Table 7. Performance of Spiral Coverage Path on Boeing 777−300ER and Airbus A321 with 100 Seats Skipped for Process C.**

| Trials | Path Length (m) | Traveling Time (s) | Total Area Covered (m²) | Coverage (%) | Avg. Speed (m²/min) |
|---|---|---|---|---|---|
| Boeing 777−300ER (Spiral) – 280.2 m² | | | | | |
| Trial 1 | 1769.8 | 15042 | 243.1 | 86.7 | 0.97 |
| Trial 2 | 1769.8 | 15421 | 241.9 | 86.3 | 0.94 |
| Trial 3 | 1769.8 | 15810 | 240.8 | 85.9 | 0.91 |
| Trial 4 | 1769.8 | 14503 | 244.6 | 87.3 | 1.01 |
| Trial 5 | 1769.8 | 14366 | 245.9 | 87.7 | 1.03 |
| Trial 6 | 1769.8 | 14972 | 243.5 | 86.9 | 0.98 |
| Trial 7 | 1769.8 | 15238 | 242.1 | 86.5 | 0.95 |
| Trial 8 | 1769.8 | 15911 | 239.7 | 85.6 | 0.90 |
| Trial 9 | 1769.8 | 15832 | 240.1 | 85.7 | 0.91 |
| Trial 10 | 1769.8 | 15390 | 241.4 | 86.1 | 0.94 |
| **Mean** | **1769.8** | **15249** | **242.3** | **86.5** | **0.95** |
| Airbus A321 (Spiral) – 225.0 m² | | | | | |
| Trial 1 | 1407.2 | 12503 | 194.5 | 86.4 | 0.93 |
| Trial 2 | 1407.2 | 12611 | 193.8 | 86.1 | 0.92 |
| Trial 3 | 1407.2 | 11894 | 196.1 | 87.1 | 0.99 |
| Trial 4 | 1407.2 | 11782 | 196.8 | 87.4 | 1.00 |
| Trial 5 | 1407.2 | 11833 | 195.9 | 87.1 | 0.99 |
| Trial 6 | 1407.2 | 11690 | 197.3 | 87.7 | 1.01 |
| Trial 7 | 1407.2 | 12390 | 194.6 | 86.5 | 0.94 |
| Trial 8 | 1407.2 | 12547 | 193.2 | 85.9 | 0.92 |
| Trial 9 | 1407.2 | 12604 | 193.1 | 85.8 | 0.92 |
| Trial 10 | 1407.2 | 12266 | 194.9 | 86.6 | 0.95 |
| **Mean** | **1407.2** | **12212** | **195.0** | **86.6** | **0.96** |

and compared against other metaheuristmic algorithms to evaluate path optimality, energy efficiency, and scalability. After Stage 2 optimization, all generated waypoints corresponding to skipped regions are successfully visited in the tested scenarios, resulting in complete coverage of all reachable target regions. Accordingly, the term CPP refers to the proposed two-stage framework as a whole, rather than to the partial sweeping behavior of Stage 1 in isolation.

Figs 15 and 16 illustrate representative trajectory outcomes for three test cases with 30 50, 100 randomly distributed seat waypoints (marked in red) with different stat/stop waypoint, respectively. In these scenarios, the robot switches between full-sized and half-sized configurations of the Smorphi robot to efficiently navigate through narrow aisles and constrained cabin geometries. The proposed method determines an optimal visitation sequence that minimizes total travel distance, thereby enhancing overall cleaning efficiency by reducing redundant motion and maintaining systematic re-cleaning coverage.

Tables 10 and 11 summarize the performance comparison between the proposed two-stage reconfigurable Coverage Path Planning (CPP) framework and the conventional fixed-size Zig-zag and Spiral strategies under aircraft cabin environments with 50 and 100 skipped seats, respectively. The evaluation considers two representative aircraft configurations, namely the Airbus A321-200 and the Boeing 777−300ER, using total cleaning time and estimated energy consumption as performance metrics.

For the 50-seat scenario, the experimental results demonstrate that the proposed two-stage GA-TSP-based method consistently outperforms both baseline strategies. Specifically, the proposed approach achieves an average reduction of

**Table 8. Performance of Zig-zag Coverage Path on Boeing 777−300ER and Airbus A321 with 100 Seats Skipped for Process C.**

| Trials | Path Length (m) | Traveling Time (s) | Total Area Covered (m²) | Coverage (%) | Avg. Speed (m²/min) |
|---|---|---|---|---|---|
| Boeing 777−300ER (Zig-zag) – 280.2 m² | | | | | |
| Trial 1 | 1663.4 | 14311 | 228.3 | 81.5 | 0.96 |
| Trial 2 | 1663.4 | 13784 | 227.6 | 81.2 | 0.99 |
| Trial 3 | 1663.4 | 13855 | 228.0 | 81.4 | 0.99 |
| Trial 4 | 1663.4 | 14726 | 227.8 | 81.3 | 0.93 |
| Trial 5 | 1663.4 | 14209 | 228.6 | 81.6 | 0.97 |
| Trial 6 | 1663.4 | 13802 | 227.9 | 81.3 | 0.99 |
| Trial 7 | 1663.4 | 14512 | 228.1 | 81.4 | 0.94 |
| Trial 8 | 1663.4 | 14001 | 227.6 | 81.2 | 0.97 |
| Trial 9 | 1663.4 | 13988 | 228.4 | 81.5 | 0.98 |
| Trial 10 | 1663.4 | 14122 | 228.2 | 81.4 | 0.97 |
| **Mean** | **1663.4** | **14113** | **228.1** | **81.4** | **0.97** |
| Airbus A321 (Zig-zag) – 225.0 m² | | | | | |
| Trial 1 | 1335.4 | 10261 | 182.1 | 81.0 | 1.07 |
| Trial 2 | 1335.4 | 10244 | 182.4 | 81.2 | 1.07 |
| Trial 3 | 1335.4 | 12103 | 181.9 | 80.9 | 0.90 |
| Trial 4 | 1335.4 | 11346 | 182.3 | 81.2 | 0.96 |
| Trial 5 | 1335.4 | 10811 | 182.6 | 81.4 | 1.01 |
| Trial 6 | 1335.4 | 11936 | 181.9 | 81.0 | 0.91 |
| Trial 7 | 1335.4 | 13912 | 182.1 | 81.1 | 0.78 |
| Trial 8 | 1335.4 | 10655 | 182.2 | 81.2 | 1.03 |
| Trial 9 | 1335.4 | 11742 | 182.3 | 81.3 | 0.93 |
| Trial 10 | 1335.4 | 11821 | 182.2 | 81.1 | 0.92 |
| **Mean** | **1335.4** | **11483** | **182.2** | **81.1** | **0.95** |

approximately 15.44% in total cleaning time and 14.15% in energy consumption compared to the single-configuration Zig-zag and Spiral methods. These gains highlight the effectiveness of combining global sweeping with localized, optimized revisitation of skipped regions.

The advantages of the proposed framework become more pronounced in the extended 100-seat scenario. As shown in Table 11, the proposed method yields substantially lower cleaning time and energy consumption for both aircraft models, with time reductions exceeding 25% and energy savings approaching 20% relative to the fixed half-size Spiral strategy. This trend indicates that the performance benefits of the reconfigurable two-stage approach scale favorably as the number of skipped or hard-to-access regions increases.

The observed improvements can be attributed to the adaptive reconfiguration mechanism and the GA-TSP-based optimization in Stage 2. In open regions such as aisles, the robot operates in its full-size configuration to maximize coverage speed and reduce traversal time. In contrast, within confined under-seat areas, the robot switches to a half-size configuration and follows an optimized waypoint sequence generated by the GA-TSP solver. This selective reconfiguration avoids unnecessary detours and redundant coverage, which are inherent to single-configuration Zig-zag and Spiral strategies. Overall, the results confirm that the proposed two-stage reconfigurable CPP framework provides a scalable and energy-efficient solution for aircraft cabin cleaning tasks involving dense and spatially distributed coverage requirements.

In addition, comparative evaluations were conducted using several well-established optimization algorithms, including Ant Colony Optimization (ACO), Differential Evolution (DE), Particle Swarm Optimization (PSO), and a baseline Greedy

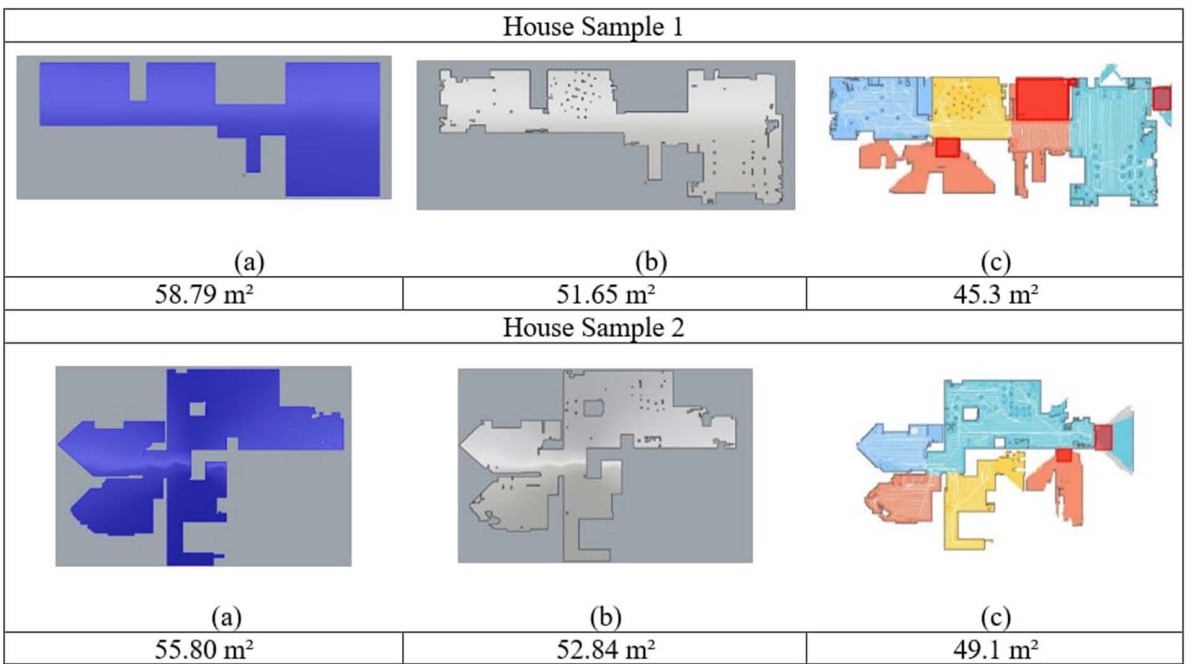

**Fig 14. Different states of Maps of the House Samples A and House Sample B environments, and their respective states throughout the analysis.** A: Original Blueprints. B: Scanned surface area. C: Travelled output from the robot travel. Both homes are in the ranges of 87.7% and 93%.

**Table 9. Comparison of Coverage, Time, and Speed in Home and Aircraft Environments.**

| Metric | Home Environments | | Aircraft Environments | | | |
|---|---|---|---|---|---|---|
| | Sample House 1 | Sample House 2 | Boeing 777−300ER | | Airbus A321-200 | |
| | | | Spiral | Zig-zag | Spiral | Zig-zag |
| Mean Coverage (%) | 87.7 | 93.0 | 86.2 | 81.4 | 86.3 | 81.2 |
| Average Cleaning Time (min) | 46.0 | 45.8 | 126.9 | 115.8 | 103.8 | 95.7 |
| Average Coverage Speed (m²/min) | 0.99 | 1.08 | 1.01 | 0.99 | 0.99 | 0.96 |

Search heuristic, to benchmark the effectiveness of the proposed GA-based approach. All benchmark algorithms were configured following standard practices in coverage path planning, with key parameters determined through preliminary tuning to ensure a fair and balanced comparison. All methods were evaluated under identical experimental conditions, including the same cabin layouts, waypoint sets, cost functions, and termination criteria, thereby ensuring consistency across comparative evaluations. A fixed computational budget was enforced by applying the same maximum number of iterations and objective function evaluations to all algorithms. Each reported result represents the average performance over 30 multiple independent runs with different random initializations, which helps mitigate stochastic bias. To assess scalability, across all evaluated scenarios, consistent performance trends are observed as the number of target seats increases from 30 to 50 and further to 100, indicating stable algorithmic behavior under increasing problem complexity and reinforcing the robustness of the comparative analysis.

- **ACO:** Number of ants = 30; pheromone evaporation rate $\rho$ = 0.4; pheromone influence $\alpha$ = 1; heuristic influence $\beta$ = 2; 100 iterations.

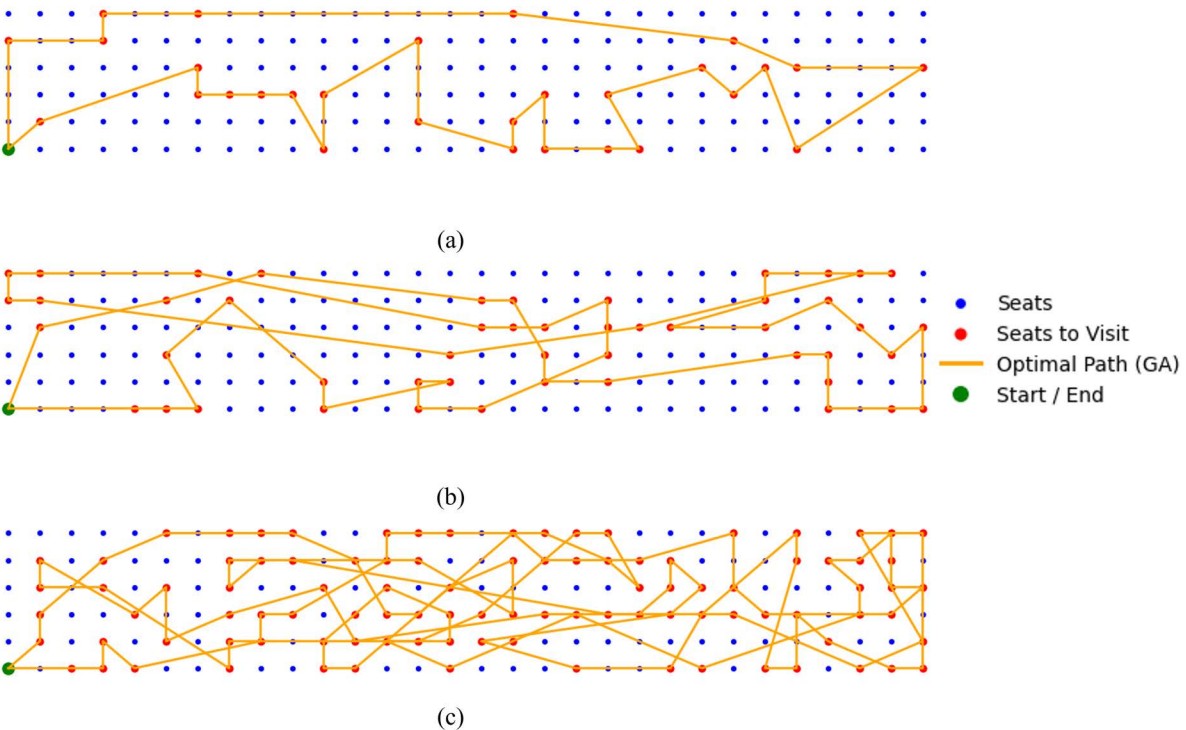

**Fig 15. Aircraft cabin with 30, 50 and 100 randomly distributed skipped seats and the corresponding optimal path generated by the proposed method, (a) 30 seats, (b) 50 seats, (c) 100 seats.**

- **DE:** Population size = 25; differential weight $F = 0.6$; crossover rate $CR = 0.9$; 100 generations.

- **PSO:** Swarm size = 30; inertia weight $\omega = 0.7$; cognitive and social coefficients $c_1 = c_2 = 1.5$; 100 iterations.

- **Greedy Search:** Sequential nearest-neighbor selection, operating deterministically without stochastic optimization.

The genetic algorithm operates on permutations of a prevalidated waypoint set rather than on continuous robot motion trajectories. All waypoints are generated exclusively from collision-free and geometrically reachable regions, taking into account the robot's active configuration and cabin constraints. As a result, the ERX crossover and swap/toggle mutation operators only alter the visiting order of feasible waypoints and cannot generate infeasible solutions involving collisions, unreachable turns, or violations of narrow aisle constraints. Feasibility is therefore preserved throughout the optimization process by construction, and no additional post-processing or feasibility repair mechanisms are required.

The numerical results in Table 12 demonstrate the outperform of the proposed GA-based optimization approach in total path length across all tested waypoint configurations, including the scenarios upto 100-seat waypoint. For all problem sizes, the proposed method consistently yields the shortest total path length compared with ACO, DE, PSO, and the Greedy Search baseline. This performance advantage is maintained as the number of target seats increases, indicating strong scalability of the GA-based optimization strategy.

As the problem size grows from 10 to 100 target seats, absolute path length reductions achieved by the proposed GA range from approximately 1.3 to 17.5 m, depending on the benchmark algorithm. Notably, the performance gap between the GA and the competing methods becomes more pronounced at larger scales, particularly in the 50- and 100-seat cases. These results highlight the GA's effectiveness in handling dense waypoint distributions and complex cabin geometries, where efficient sequencing plays a critical role in minimizing total travel distance.

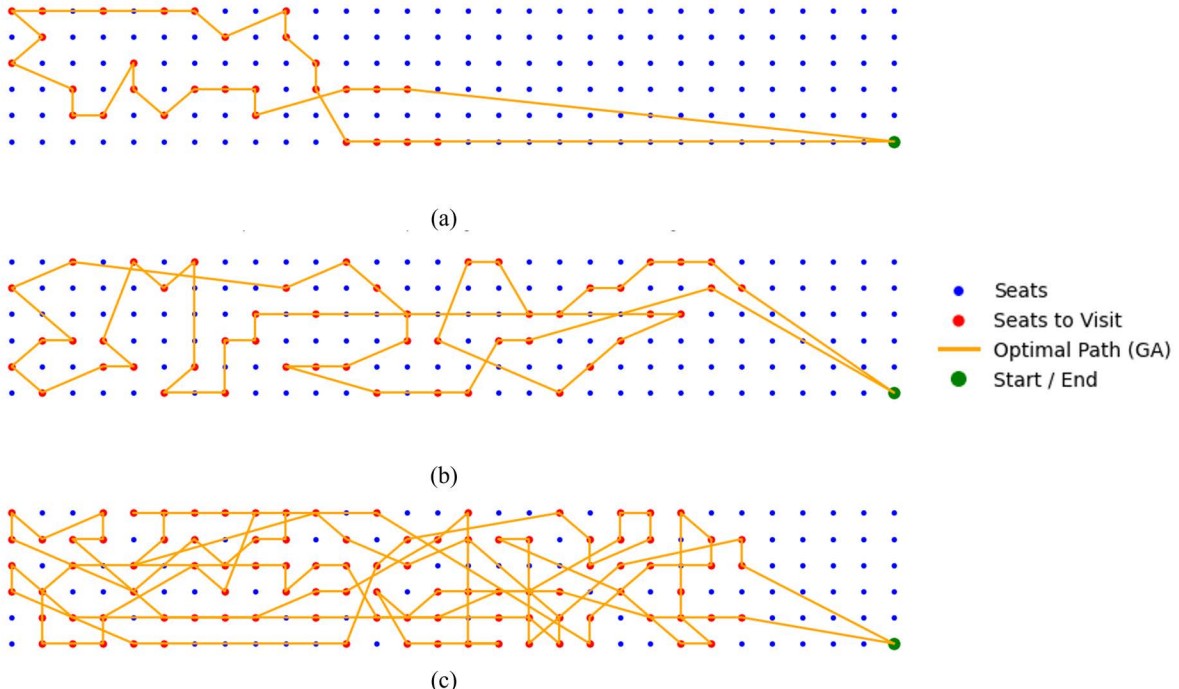

(a)

(b)

(c)

**Fig 16. Aircraft cabin with 30, 50, 100 randomly distributed skipped seats and the optimal path generated by the proposed GA-based method with different starting and ending locations, (a) 30 seats, (b) 50 seats, (c) 100 seats.**

**Table 10. Comparison of Cleaning Time and Energy Consumption for Different Coverage Strategies in Aircraft Cabin Cleaning (50-Seat Scenario).**

| Aircraft Type | Proposed (GA-TSP) | | Zigzag (Half-size) | | Spiral (Half-size) | |
|---|---|---|---|---|---|---|
| | Time (s) | Energy (Wh) | Time (s) | Energy (Wh) | Time (s) | Energy (Wh) |
| A321-200 | 412.1 | 340.6 | 487.2 | 403.3 | 501.0 | 425.2 |
| 777−300ER | 585.1 | 478.0 | 692.2 | 556.3 | 728.3 | 594.4 |
| **Reduction (%)** | 15.44 | 14.15 | – | – | – | – |

**Table 11. Comparison of Cleaning Time and Energy Consumption for Different Coverage Strategies in Aircraft Cabin Cleaning (100-Seat Scenario).**

| Aircraft Type | Proposed (GA-TSP) | | Zig-zag (Half-size) | | Spiral (Half-size) | |
|---|---|---|---|---|---|---|
| | Time (s) | Energy (Wh) | Time (s) | Energy (Wh) | Time (s) | Energy (Wh) |
| A321-200 | 823.4 | 667.1 | 11483 | 667.7 | 12212 | 703.6 |
| 777−300ER | 1168.2 | 742.6 | 14113 | 831.7 | 15249 | 884.9 |
| **Reduction (%)** | 28.3 | 18.6 | – | – | – | – |

Together with the total path length reported in Table 12, Table 13 summarizes the estimated energy consumption associated with each optimization algorithm. Note that energy values are computed using the total cost function $T_c$ as in Eq 8. As one can observe the consistent with the path length results, the proposed GA-based method achieves the lowest energy consumption across all tested scenarios, form 10–100-seat case. Compared with other metaheuristic approaches

**Table 12. Comparison of Total Path Length (m) for Different Algorithms with Varying Numbers of Target Seats.**

| # Target Seats | Proposed GA | ACO | DE | PSO | Greedy Search |
|---|---|---|---|---|---|
| 10 | 15.2 | 16.8 | 17.1 | 16.5 | 18.0 |
| 20 | 28.4 | 30.1 | 31.3 | 29.5 | 32.0 |
| 30 | 41.6 | 44.2 | 45.0 | 43.1 | 46.5 |
| 40 | 54.8 | 58.5 | 59.7 | 56.2 | 61.0 |
| 50 | 67.9 | 72.3 | 73.8 | 69.7 | 75.5 |
| 100 | 131.4 | 140.6 | 144.2 | 136.8 | 148.9 |

**Table 13. Comparison of Estimated Energy Consumption (Wh) for Different Algorithms with Varying Numbers of Target Seats.**

| # Target Seats | Proposed GA | ACO | DE | PSO | Greedy Search |
|---|---|---|---|---|---|
| 10 | 76.3 | 84.7 | 86.4 | 83.6 | 90.8 |
| 20 | 142.5 | 151.9 | 157.6 | 148.3 | 160.4 |
| 30 | 208.2 | 221.6 | 225.9 | 216.7 | 233.5 |
| 40 | 274.8 | 293.4 | 299.2 | 281.5 | 305.9 |
| 50 | 340.6 | 362.8 | 369.3 | 349.7 | 378.4 |
| 100 | 667.1 | 704.5 | 721.3 | 689.8 | 742.6 |

(ACO, DE, and PSO), the GA reduces total energy usage by approximately 8–15%, while achieving reductions of 14–18% relative to the Greedy Search baseline. These energy savings increase with the number of target seats, confirming that the GA-based optimization maintains its efficiency advantage as task complexity and waypoint density grow. Overall, the results demonstrate that the proposed approach offers a scalable and energy-efficient solution for CCPP in large-scale, geometrically constrained aircraft cabin environments.

In summary, the proposed GA-based TSP optimization framework consistently outperforms conventional and state-of-the-art algorithms in terms of travel distance, energy efficiency, and scalability. Its adaptability to different cabin layouts and varying target densities demonstrates its strong generalization capability and makes it a promising approach for intelligent robotic cleaning operations in structured yet constrained environments. This improvement can be attributed to the design of the GA framework, which incorporates domain-specific crossover and mutation operators that preserve spatial adjacency among waypoints, effectively minimizing unnecessary detours. Additionally, the fitness function integrates a penalty term to discourage backtracking and abrupt turning angles, leading to smoother and more feasible motion trajectories for real-world robot execution. While ACO and PSO offer rapid convergence, they often become trapped in local optima as waypoint density increases. DE performs adequately for mid-scale problems but exhibits reduced scalability for larger instances. The Greedy TSP-CPP heuristic, though computationally lightweight, lacks global optimization capability, resulting in longer travel paths and higher redundancy.

## 5 Discussion and limitations

This study proposes a two-stage CPP framework for aircraft cabin cleaning using a reconfigurable mobile robot. The framework combines a fast global sweeping strategy (Stage 1) with a waypoint-based optimization process (Stage 2) to achieve complete coverage of all reachable under-seat regions while maintaining computational efficiency.

### 5.1 Discussion

A key characteristic of the proposed framework is that coverage completeness is guaranteed at the framework level rather than at an individual stage. Stage 1 is intentionally designed as a rapid and conservative sweeping process under

the robot's full configuration, prioritizing efficiency and safety over exhaustive coverage. As a result, partial coverage rates (81–86%) are expected at this stage, particularly in narrow or obstructed under-seat areas. All skipped or uncertain regions are explicitly logged and transformed into mandatory waypoints, which are revisited in Stage 2. The extended experimental results for 30, 50, and 100 target seats confirm that all generated waypoints are successfully visited, resulting in complete coverage of all reachable target regions.

The waypoint-based formulation enables flexible scaling of the optimization problem. Although the aircraft cabins considered in this study contain more than 200 physical seats, the number of target seats optimized in Stage 2 corresponds to the effective set of skipped or hard-to-access regions identified after Stage 1, rather than the total number of seats. This explains why earlier experiments focused on up to 50 target seats. Additional experiments with 100 target seats further demonstrate that the proposed GA-based optimization does not exhibit performance degradation as problem size increases, maintaining consistent advantages over ACO, DE, PSO, and greedy baselines in terms of path length and estimated energy consumption.

### 5.2 Modeling assumptions and limitations

Several simplifying assumptions are adopted in this work to maintain a clear focus on planning-level feasibility and scalability. First, energy consumption is estimated using a distance-based approximation to enable fair and consistent comparison among optimization algorithms. Dynamic effects such as turning effort, acceleration and deceleration, frictional losses, and corrective maneuvers in narrow passages are not explicitly modeled. Consequently, the reported energy values should be interpreted as relative performance indicators rather than absolute physical measurements, and no explicit upper bound on the estimation error is provided. Incorporating fine-grained cabin surface properties and execution-level interaction effects would be necessary for deployment-oriented validation.

Second, In this work, coverage is defined in terms of geometric traversal of target regions by the robot, consistent with standard coverage path planning formulations, rather than in terms of cleaning effectiveness or post-cleaning verification. Cleaning effectiveness factors such as coverage width, overlap rate, edge effects, suction efficiency, or sterilization verification are not explicitly considered. The current implementation assumes that traversal of a target region implies effective cleaning within the robot's operational footprint. Incorporating mechanism-specific coverage width, overlap constraints, and cleaning effectiveness verification represents an important direction for future work.

Third, the framework is evaluated as an offline planning approach based on a reconstructed static cabin map. Dynamic changes caused by cleaning staff movement, debris shifts, or seat recline are not modeled. While the waypoint-based formulation allows newly detected regions to be appended and locally re-optimized, full online replanning with guaranteed latency and complexity bounds is beyond the scope of this study.

Finally, the simulation environment models macro-level cabin geometry derived from aircraft layout diagrams, including seat arrangement, aisle width, and under-seat clearances. Micro-scale surface variations such as carpet texture, floor thresholds, or metal seat rails are not explicitly represented. Therefore, the conclusions of this work are limited to planning-level behavior rather than execution-level performance in real cabins.

## 6 Conclusion and future work

This paper presented a two-stage cleaning framework for autonomous operations within aircraft cabin interiors using the reconfigurable Smorphi robot. The study demonstrated the feasibility of employing both full-size and half-size robotic configurations for effective coverage and localized re-cleaning within a highly structured and cluttered environment. The results verified that the proposed Coverage Path Planning (CPP) framework—comprising deterministic global coverage and GA-based local path optimization—can efficiently handle the spatial constraints and cleaning requirements of modern aircraft cabins.

From a feasibility standpoint, this work provides an important first step toward realizing autonomous robotic cleaning systems tailored for confined public environments such as aircraft cabins. The experiments showed that preprogrammed and adaptive path planning strategies are essential for achieving complete coverage and reducing redundant motions, particularly in scenarios constrained by fixed seat layouts and narrow aisles. The outcomes suggest significant potential for robotic cleaning in safety-critical and hygiene-sensitive environments, such as during pandemic conditions.

Future research will focus on several directions to enhance the system's overall performance, robustness, and scalability.

### 6.1 Robot coordination for time-constrained operations

To meet airline turnaround time requirements—typically for the Airbus A321-200 (Air Seoul) and for the Boeing 777−300ER (Singapore Airlines)—future work will investigate multi-robot coordination strategies. By partitioning the cleaning task among multiple robots, the system can significantly reduce total operation time while maintaining full coverage. This concept aligns with recent studies on cooperative multi-robot cleaning [51]. Preliminary simulations under worst-case conditions indicate that multi-robot deployment can reduce cleaning duration proportionally to the number of coordinated agents.

### 6.2 Hybrid coverage path planning for enhanced efficiency

Analysis of the current CPP strategies suggests that neither a full spiral pattern nor a full zigzag pattern provides optimal performance across all cabin areas. The zigzag approach offers consistent coverage in open regions such as aisles, while the spiral pattern demonstrates superior performance under seats and within confined spaces such as galleys or crew areas. Therefore, future work will explore a hybrid pattern that combines the advantages of both methods—using a spiral pattern for local cluttered areas and a zigzag approach for linear sections—to balance coverage completeness, cleaning duration, and maneuver efficiency.

### 6.3 Shape-reconfigurable robotic design

Although the *Smorphi* robot can switch between full-size and half-size configurations, certain areas—such as deep spaces between seats and near fuselage walls—remain inaccessible. To overcome these physical constraints, future research will investigate more flexible, shape-shifting robotic architectures [52]. Such reconfigurable designs would allow the robot to dynamically alter its geometry to adapt to irregular and constrained spaces, improving accessibility and coverage performance. Integrating this capability with the proposed CPP and adaptive local re-cleaning framework would significantly enhance the robot's ability to operate autonomously in complex cabin environments.

### 6.4 Online replanning under dynamic cabin conditions

The current framework is evaluated as an offline planning approach based on a reconstructed static cabin map. In real-world deployments, however, cabin conditions may change due to cleaning staff movement, debris shifts, or seat recline. Future work will focus on incorporating onboard sensing and real-time perception to support online replanning. In particular, uncertainty-aware detection and incremental waypoint updates could enable the robot to dynamically adjust its plan while preserving coverage guarantees. Developing bounded-time replanning strategies to ensure computational tractability and latency guarantees remains an important research challenge.

### 6.5 Mechanism-aware coverage and integrated sterilization

The present study assumes a vacuum-based cleaning mechanism and defines coverage in terms of geometric traversal of target regions. Extending the framework to support integrated sterilization mechanisms, such as UV irradiation or

spray-based disinfection, will require modeling additional constraints related to exposure dosage, line-of-sight obstruction, overlap requirements, and safety distances. Future work will explore mechanism-aware coverage formulations by refining waypoint generation density and augmenting the cost function to explicitly account for sterilization effectiveness and safety constraints. Incorporating mechanism-specific coverage width, overlap constraints, and cleaning effectiveness verification represents an important direction for future work.

### 6.6 High-fidelity energy modeling and parameter sensitivity

The distance-based energy approximation adopted in this work enables efficient algorithm comparison but does not explicitly capture dynamic effects such as turning effort, acceleration, friction, or corrective maneuvers. Future research will investigate higher-fidelity energy models that incorporate these factors, supported by empirical validation on physical robotic platforms. In addition, systematic sensitivity analysis of cost-function weighting parameters will be conducted to improve robustness and facilitate parameter tuning across different cabin configurations and task requirements. Several simplifying assumptions are adopted in this work to maintain a clear focus on planning-level feasibility and scalability. First, energy consumption is estimated using a distance-based approximation to enable fair and consistent comparison among optimization algorithms. Dynamic effects such as turning effort, acceleration and deceleration, frictional losses, and corrective maneuvers in narrow passages are not explicitly modeled. Consequently, the reported energy values should be interpreted as relative performance indicators rather than absolute physical measurements, and no explicit upper bound on the estimation error is provided. Incorporating fine-grained cabin surface properties and execution-level interaction effects would be necessary for deployment-oriented validation. Extending the simulation to incorporate such factors, coupling the proposed planner with low-level motion controllers, and conducting comprehensive statistical benchmarking studies and integrating continuous motion-level validation and dynamic collision avoidance within the GA optimization loop constitutes a natural extension are identified as important directions for future work.

## Supporting information

**S1 File. Dataset for Tables.**
(ZIP)

## Acknowledgments

The authors thank Dr. Prabakaran Veerajagadheswar for his insightful advice during the revision of the paper.

## Author contributions

**Conceptualization:** Cong Hien Dinh, Chong Yong Qi, Anh Vu Le.

**Data curation:** Cong Hien Dinh.

**Formal analysis:** Anh Vu Le.

**Methodology:** Chong Yong Qi, Guangming Chen, Anh Vu Le.

**Software:** Chong Yong Qi.

**Supervision:** Huynh Van Van, Rajesh Elara Mohan.

**Validation:** Rajesh Elara Mohan, Anh Vu Le.

**Visualization:** Guangming Chen.

**Writing – original draft:** Cong Hien Dinh, Chong Yong Qi.

**Writing – review & editing:** Huynh Van Van, Guangming Chen, Rajesh Elara Mohan, Anh Vu Le.

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
