## [Decision Letter · Decision Letter 0]

15 Jan 2026

PONE-D-25-61185Genetic Algorithm-Based Coverage Path Planning for Autonomous Aircraft Cabin Cleaning by Reconfigurable RobotPLOS One

Dear Dr. Le,

Thank you for submitting your manuscript to PLOS ONE. After careful consideration, we feel that it has merit but does not fully meet PLOS ONE’s publication criteria as it currently stands. Therefore, we invite you to submit a revised version of the manuscript that addresses the points raised during the review process.

More specifically, in the revised version of the paper please clarify and justify the algorithmic design (Stage-1 detection, waypoint generation, cost weights, feasibility of GA paths), report full final coverage after Stage-2, and strengthen the experimental validation (fair multi-run comparisons, scalability beyond 50 seats, and realistic motion/coverage modeling). In addition, please explain why only up to 50 seats are tested and ensure formatting, figures, and references are fully consistent.

We look forward to receiving your revised manuscript.

Kind regards,

Camelia Delcea

Academic Editor

PLOS One

Journal Requirements:

3. We note that your Data Availability Statement is currently as follows: All relevant data are within the manuscript and in Supporting Information files.

4. We note that Figures 1,2 and 5 in your submission contain copyrighted images. All PLOS content is published under the Creative Commons Attribution License (CC BY 4.0), which means that the manuscript, images, and Supporting Information files will be freely available online, and any third party is permitted to access, download, copy, distribute, and use these materials in any way, even commercially, with proper attribution. For more information, see our copyright guidelines: http://journals.plos.org/plosone/s/licenses-and-copyright.

1. You may seek permission from the original copyright holder of Figures 1,2 and 5 to publish the content specifically under the CC BY 4.0 license.

Reviewers' comments:

Reviewer's Responses to Questions

**Comments to the Author**

1. Is the manuscript technically sound, and do the data support the conclusions?

Reviewer #1: Yes

Reviewer #2: Yes

2. Has the statistical analysis been performed appropriately and rigorously?

Reviewer #1: Yes

Reviewer #2: Yes

3. Have the authors made all data underlying the findings in their manuscript fully available?

Reviewer #1: Yes

Reviewer #2: Yes

4. Is the manuscript presented in an intelligible fashion and written in standard English?

Reviewer #1: Yes

Reviewer #2: Yes

5. Review Comments to the Author

Reviewer #1: 1. There are above 200 seats in the aircraft cabin of Boeing 777-300ER and Airbus A321-200 respectively, but why the maximum "Numbers of Target Seats" is 50 in the table 9 and table 10? Please explain it briefly in the relative sections of this manuscript.

2. Please maintain consistency in the text font, size, and formatting of the images and tables throughout the manuscript.

3. Please maintain consistency in the reference format, such as: 1) DOI number, either all references are labeled or none are labeled, unless some references do not have a DOI number. 2) The capitalization, full spelling/simplified spelling, etc. of the author's name.

Reviewer #2: 1. What triggers the skipping of seat rows in Stage 1? How exactly are “hard-to-access seat rows” detected and determined? What are the thresholds, sensor types, and false positive rates? If false positives cause missed scans, Stage 2 cannot compensate.

2. How is the waypoint set W generated? Is one point assigned per row? Or are multiple points used to cover the area beneath seats? How are waypoints mapped to the actual coverage area?

3. How are γ1, γ2, and γ3 determined in the cost function (8)? Has sensitivity analysis been conducted? Does increasing γ3 cause the algorithm to favor fewer reconstructions at the expense of coverage?

4. Emove uses a linear distance model (α·d/β·d), yet robot energy consumption heavily depends on turning, acceleration/deceleration, ground friction, and course correction in narrow passages. Why are these factors neglected? Is an upper bound on error provided?

5. Stage 1 coverage is only 81–86%. What is the final coverage rate after Stage 2? Does it meet the definition of “complete coverage”? If not, is the CCPP naming accurate?

6. Is the GA vs. ACO/DE/PSO comparison fair? Were all algorithms evaluated under the same computational budget (time/iterations/function evaluations)? Were results repeated ≥30 times to provide mean ± standard deviation with significance testing?

7. Table 9 only shows results for 50 target points, but the text mentions 10–100 seats. Where are the results for 100 seats? Does the algorithm degrade at larger scales?

8. GA uses ERX crossover with swap/toggle mutation: Could this generate infeasible paths (collisions/unreachable turns/violating narrow aisle constraints)? How is feasibility ensured?

9. Aircraft cleaning efficiency depends not only on path planning but also on cleaning coverage width, cleaning mechanisms, and cleaning effectiveness verification. Does the coverage definition in the paper account for cleaning width/overlap rate/edge missed scans?

10. If cabin conditions change due to cleaning staff movement, debris shifts, or reclined seats, this alters the map and navigability. Does the framework support online replanning? Or is it entirely offline? If online, how are complexity and latency guaranteed?

11. The paper notes a “lack of integrated sterilization capabilities.” Does the current method only apply to vacuuming? If UV/spray disinfection is added, coverage and safety constraints (obstruction, dosage) will significantly alter CPP design. Is the framework extensible?

12. Maps for both aircraft models are sourced from manuals/layout diagrams. However, real cabins exhibit detailed variations (carpets/thresholds/metal rails). Are these modeled in simulations? If not, could conclusions be over-extended?

6. PLOS authors have the option to publish the peer review history of their article (what does this mean?). If published, this will include your full peer review and any attached files.

Reviewer #1: **Yes:** Xiaoyou Yu

Reviewer #2: No

---

## [Author Response · Author response to Decision Letter 1]

11 Mar 2026

Thank you for allowing us to revise the manuscripts. We have addressed the point-to-point comments of the editors and the reviewers. Please find the uploaded revised manuscript and response for reviewers file for your details.

---

## [Decision Letter · Decision Letter 1]

30 Apr 2026

Genetic Algorithm-Based Coverage Path Planning for Autonomous Aircraft Cabin Cleaning by Reconfigurable Robot

PONE-D-25-61185R1

Dear Dr. Le,

We’re pleased to inform you that your manuscript has been judged scientifically suitable for publication and will be formally accepted for publication once it meets all outstanding technical requirements.

Kind regards,

Camelia Delcea

Academic Editor

PLOS One

Additional Editor Comments (optional):

Reviewers' comments:

Reviewer's Responses to Questions

**Comments to the Author**

1. If the authors have adequately addressed your comments raised in a previous round of review and you feel that this manuscript is now acceptable for publication, you may indicate that here to bypass the “Comments to the Author” section, enter your conflict of interest statement in the “Confidential to Editor” section, and submit your "Accept" recommendation.

Reviewer #1: All comments have been addressed

2. Is the manuscript technically sound, and do the data support the conclusions?

Reviewer #1: Yes

3. Has the statistical analysis been performed appropriately and rigorously?

Reviewer #1: Yes

4. Have the authors made all data underlying the findings in their manuscript fully available?

Reviewer #1: Yes

5. Is the manuscript presented in an intelligible fashion and written in standard English?

Reviewer #1: Yes

6. Review Comments to the Author

Reviewer #1: The authors have adequately addressed comments raised in a previous round of review, and I feel that this manuscript is now acceptable for publication.

7. PLOS authors have the option to publish the peer review history of their article (what does this mean?). If published, this will include your full peer review and any attached files.

Reviewer #1: **Yes:** Xiaoyou Yu

---

## [Editor Report · Acceptance letter]

PONE-D-25-61185R1

PLOS One

Dear Dr. Le,

I'm pleased to inform you that your manuscript has been deemed suitable for publication in PLOS One. Congratulations! Your manuscript is now being handed over to our production team.

Kind regards,

on behalf of

Dr. Camelia Delcea

Academic Editor

PLOS One